# A Cross-Cultural Analysis for Plastic Waste Perception of Students from Romania and Turkey

Graţiela Dana Boca [1,*], Arzum Işitan [2], Evren Çağlarer [3] and Sinan Saraçli [4]

1　Department of Economics, Faculty of Sciences, Technical University of Cluj Napoca, 430122 Baia Mare, Romania
2　Department of Mechanical Engineering, Faculty of Technology, Pamukkale University, Kinikli Kampus, 20160 Denizli, Turkey; aisitan@pau.edu.tr
3　Department of Mechatronic, Technology Faculty, Kırklareli University, 39100 Kırklareli, Turkey; ecaglarer@gmail.com
4　Department of Biostatistics, Faculty of Medicine, Balıkesir University, 10145 Balıkesir, Turkey; ssaracli@balikesir.edu.tr
*　Correspondence: gratiela.boca@econ.utcluj.ro

**Abstract:** The article brings to attention a cross cultural model related to the perception of students in relation to the current problem of plastic waste. To create the model, a questionnaire was applied online in two countries at the same time, among students from different specializations. The survey was structured in several parts, with the first part meant to identify individual characteristics of the responders, the second part to identify their knowledge about plastic, determine their beliefs in the new material—bioplastic, their preference in using plastic or bioplastic, and the last part meant to determine students' attitude towards the environment. The model wants to highlight the preferences and knowledge of students about plastic, the degree of information and students' knowledge about plastic waste, and if these are influenced by culture; in our case, the country was considered. Also, we established that gender or specialization have no influence on the perception of bioplastic. A total of 39.79% of the students from both countries participate in and attend conferences about nature protection and plastic waste, and only 58.69% of the students do not participate in any conferences about nature conservation or recycling materials. As a conclusion, we can mention that Turkish students are more responsible and more active in environmental activities regarding plastic waste in comparison with Romanian students. In comparison with Romanian students, Turkish students are more careful when it comes to recycling waste plastic and when choosing products that are less harmful to nature. The young generation is open to selective recycling, even if they sometimes do not follow the established rules. Based on this model, common problems can be identified and universities, as incubators of ideas, can welcome the use of the necessary methods and tools to stimulate care and students' awareness of the environment and its protection.

**Keywords:** plastic waste; recycling; knowledge; perception; beliefs; cross cultural model



## 1. Introduction

Plastic is probably the biggest challenge today, but there are problems with all types of packaging waste, so the population needs a favorable framework through which selective collection can no longer be considered an effort. Romania ranks last in the European Union in terms of packaging waste recycling, the percentage falling in 2020 to 39%, with almost 5 percentage points less than in 2019, according to the latest European statistics cited by Clean Recycle [1]. Comparatively, the EU champions in this chapter are Belgium with 79%, the Netherlands with 74%, and Luxembourg with 72%. The environmental targets have increased in 2023, so that packaging waste should be recycled in a proportion of 65%. From 2025, this percentage will rise to 70% [1].

Romania is faced with the unpredictability of the legislation, with a lack of awareness and education of the population towards eco-responsible behavior, but also with the lack of infrastructure [2]. Jambeck et al. [3] and Karasik [4] mention in their research that Turkey has one of the highest volumes of both plastic and overall waste in the world, with a significant waste footprint in the Mediterranean Sea, with the largest mass of mismanaged plastic waste. Edelson et al. [5] mentioned that plastic pollution is not only limited to its recycling and selection but also to the management of plastic and pollution, which must be improved. Karasik [4] emphasized that a solution to the problem of plastic pollution would be the promotion and raising awareness of this problem, which can have a positive impact on students and on the innovation of new effective solutions. The use of plastic is one of the most pressing environmental issues facing humanity in order to reduce global warming. A large part of this waste corresponds to the food industry and its packaging.

When we use the word "plastic" we generally mean to describe the multitude of shapes and forms in which this material appears. There are seven types of plastic that vary in chemical composition, purpose, recyclability, and hazardous nature. Regardless of which category of the seven types of plastic they fall into, all plastics must be recycled or reused to move towards a circular economy. We must also mention other types of plastic materials, derived contaminants, such as the emerging contaminants BPA, 4-noniphenol, PFOS, micro plastics, heavy metals, and many others. Plastic must be recycled or reused for a circular economy to mitigate pollution and its impact on the planet. Most recycling programs do not accept the seventh category of plastics because they are difficult to identify and separate for recycling.

Reducing plastic consumption is extremely essential to mitigating pollution and its impact on the planet. PET plastic is mainly used as packaging for juice, water, medicine jars, household cleaning products, and more. PET plastic is one of the most frequently recycled. The use of plastic is one of the most pressing environmental issues facing humanity in order to reduce global warming. Also a large part of this plastic waste corresponds to the food industry and its packaging. Therefore, it is important to know which packaging is dangerous and which safe options exist. The risks to human health and the environment associated with the use of plastic containers are huge. Aurisano et al. [6] specified, however, that the activation of a circular economy for plastics in Europe is an ambitious objective. Assessing their impact on the environment and on human health throughout the life cycle of plastic products is paramount. They identified 1518 chemicals of concern related to plastic, replacing them with safer alternatives in support of a circular plastics economy. Verla et al. [7], however, consider that the current problem faced by researchers globally is micro plastic, as well as the toxic chemical pollution of the ecosystem and the impact of ingested micro plastic on human health.

## 2. Literature Review on Plastic Waste

Kaushik Dowarah et al. [8] study the attitude towards plastic and pollution among students in India, which can have a positive impact and lead to increased student awareness and the stimulation of efficient innovative solutions. Hammami et al. [9] carried out a survey but having as its target high school students and, from his research, observed that most students understand from an early age how harmful plastic waste is for the environment, but also the level of knowledge about the environment demonstrated the need for strategies to address deficiencies, to stimulate change with government support, and to increase the opportunities to adopt pro-environmental behaviors. Singh Chauhan and Punia [10] and Singh Chauhan et al. [11] also focused on the problem of plastic pollution, which is currently a global problem but which is not yet understood. That is why the authors propose the integration of issues related to plastic materials into the education system for environmental awareness. Even if teachers and students are involved, they noticed that increased environmental programs regarding management and impact practices are needed.

Auld [12] also conducted a study on students from Shayla University and shows a wide range of consumption habits with different consequences for the environment, disposal, and misuse of waste. It appears that key tools and information are needed for students to consume and use plastic responsibly. Most were found to be aware of the consequences of plastic pollution, but still many were uneducated about proper usage and disposal methods on campus.

Izzah Abd Hamid et al. [13] emphasized the importance of continuously educating the younger generation on the importance of pollution and environmental conservation. The results indicated that the respondents' behavior was still dependent on plastic products. Awareness behavior among future generations is important to reduce the use of plastic and conserve nature.

Jariyah et al. [14] analyzed in their study the unsolved problems in Indonesia, which so far is that of waste. They pointed out that, for now, the community is still lacking awareness about this issue. Students became aware of the use of water bottles as an effort to reduce the use of plastic bottles. Many studies have linked education to environmental awareness and behavior, such as Situmorang et al. [15], who consider that e-Environmental education at the academic level and the awareness of students about the problems related to plastic waste is a priority in Taiwan as well. Aikowe and Mazancová [16] noted that a differentiated approach is needed, especially among students, because of the impact environmental education can have on pro-environmental awareness. Depending on the country in question, different impacts can be observed, while promoting pro-environmental behaviors is also different. In his work, Aikowe [17] took into account the influence of social norms, namely plastic recycling, and promotes the notion of sustainable education in higher education institutions in Nigeria. Another recommendation was for policy makers and universities to be urged to take proactive steps to adapt ecological volunteering curricula and activities through university educational policies.

Araya Wongklaw et al. [18] focused on single-use plastic pollution, which has become one of the biggest environmental problems. Although people understand the harmful effects of plastic pollution, most continue to use single-use plastic products. The study aimed to understand the relationship between students' behaviors, knowledge, and awareness in reducing global warming caused by excessive use of plastic products. Results showed that there is a significant correlation between the level of awareness of plastic pollution and behavior change, i.e., trying to avoid and reduce the use of single-use plastic bags or stop using plastic straws.

There are still unsolved problems regarding plastic waste; one factor would be that the community still lacks awareness about this issue. Many studies have linked education to environmental awareness and behavior, such as Situmorang et al. [15], who consider that e-Environmental education at the academic level and the awareness of students about the problems related to plastic waste are and must be priorities in Taiwan as well.

Pariyar et al. [19] and later Ferdous and Das [20] carried out a study on the attitude of students towards the use of plastic, understanding the use of plastic materials, and their influence on human health. The transfer of knowledge into behavior has been affected by educational barriers and other societal factors. Kaur Simarjeet and Jeganathan [21] made the same observation that plastic is an extremely useful material, but that it also brings about certain plastic-use related health hazards among adolescent girls. The results of the study revealed that 52% of the respondents had poor knowledge, 48% had average knowledge, and none of them had good knowledge about the health hazards of using plastic.

There is a large body of research on environmental education programs. Ryan Collin [22] studied the impact on students in evaluating the effectiveness of these programs. Collin [22] aims to evaluate an environmental education program on the sustainability of plastics; students were informed about the variability of the properties of plastics and the sorting and reshaping processes of recycling. Another aspect highlighted by Collin [22] was building pro-environmental attitudes and behaviors and civic responsibility regarding

plastic waste. In conclusion so far, at high school, college, or university level it is necessary to measure the attitude, beliefs, and knowledge about plastic waste and preference towards recycling and new biodegradable plastics to identify how they can be motivated to recycle plastic and raw material.

Loveth Aikowe [17] gives special attention to plastic recycling amongst students, Qu et al. [23] focus on waste separation and recovery on campus, Wang et al. [24] on students' intentions to use recyclable packaging. Thus, they all bring a new vision regarding the role of universities and the importance of sharing environmental problems, especially regarding plastic waste.

Harman and Yenikalayci [25] in their research aimed to determine the level of awareness of science students regarding waste management. A significant proportion of students were aware of the effects of education, research, and project activities on recovery, reuse, recycling, use of plastic bags, and zero waste practices in waste management, as well as the place and importance of the individual in waste management.

Olukunle [26], in his research on the perception and behavior of students on waste management, concludes that most students do not know the basis of waste management practices such as types of waste, segregation of waste at source according to their types, and disposal of waste in baskets accordingly. Uehara et al. [27] appreciate the knowledge of the students, the fact they know the rules so as to effectively improve the separation of plastic waste on campus for a more circular economy.

A new term, namely, the "plastic era", was introduced by Van Rensburg et al. [28]. Green products are considered to resolve global plastic pollution, as Moshood et al. [29,30] mention in their research, highlighting that the period for biodegradable plastic brings a new period of innovations in that field.

Fadhullah et al. [31] and Henderson and Dumbili [32] in their research found that, for many young Nigerians, the perception of plastic waste from the social point of view is that it is considered to be cool to use trash cans or recycling, which implies the need for individual responsibility. All these represent a sign of modernity and increased social status. Also Gherheş et al. [33] investigated students' perceptions regarding plastic waste, and the results present that the majority of students still need to be familiarized with plastic waste through different campaigns, trainings, courses, etc.

Rosario and Dell [34] were concerned about the environment and the importance of sustainable materials in industry. They consider materials and laboratory courses allow students to test the biodegradability of plastic materials so that students understand the new materials. Thus the implementation of biodegradable testing in a curriculum provides active learning through practice tests and encourages students to engage in lifelong learning in order to continue to develop their knowledge of emerging materials.

Currently, the recycling of metal, wood, paper, and especially all cardboard packaging, has improved significantly. On the other hand, as regards plastic materials, their recycling and removal from the market has not yet been resolved. The European Union has implemented various regulations regarding packaging and packaging waste recycling and the market implementation of eco-sustainable packaging. Rossi et al. [35] have developed an innovative and sustainable composite material for packaging, i.e., a new eco-friendly material based on the combination of natural biodegradable fibers and biopolymers consisting of straw and biodegradable plastic. The authors present the results of the new material, showing a good match with the characteristics of current polymers, suggesting that this material can be used as a potential substitute in packaging applications. Also, Fiorineschi et al. [36] in their paper present the application of a systematic engineering design procedure also adapted for eco-friendly production of compostable straw fiber packaging and bioplastic but in the field of viticulture (the obtained boxes are intended to be used for wine bottles).

Olteanu and Gorghiu [37] argue that scientific actions through investigations, experiments, and research are important in attracting the younger generation to science. Important subjects are promoted to students in this sense, the subject of biodegradability

of plastic materials, in various approaches, addressing the problems of our day, answering scientific questions, or trying to do so. Olteanu and Gheorgiu [37] emphasized that the involvement of students in research leads to an increase in the interest of the young generation in science. The results obtained through specific approaches to STEM (Science, Technology, Engineering, Math) education led students to become aware of this sensitive issue. The assessment of students' interest in science after the implementation of the scientific actions showed their powerful confidence in science, being ready to participate in or benefit from collaborative scientific projects, the support of their families who believe that the understanding and knowledge of science is useful for the whole life. Also when assessing students' interest in science, students provided positive feedback related to teachers' ability.

The transition to a circular and sustainable economy can be viewed from a socio-technical point of view response to environmental impact. Bioplastics, typically plastics made from bio-based polymers, should contribute to more sustainable plastic life cycles as part of a circular economy. Ali et al. [38] in their work carry out a review that highlights the harmful effects of fossil-based plastic on the environment and human health, as well as the massive need for green alternatives such as biodegradable ones and bioplastics.

The use of the new types of bioplastics derived from renewable resources and choosing the appropriate end-of-life option may be the right direction to ensure the sustainability of bioplastic production. At the same time, clear regulations are required and financial incentives to scale up with application in the market having a truly lasting impact.

## 2.1. Cross-Cultural Model Regarding Plastic Waste

The idea of the cross-cultural model, between two different cultures, has been implemented by many authors from different countries, especially among young people, to examine the differences of attitude, behavior, and perception regarding plastic waste.

Because the plastic waste and impact upon the environment remain yet unsolved problems, Lorenzo et al. [39] have created a model in the field of ecotourism, comparing ecotourists from two different cultures, Chilean and Spanish. The results indicate that the proposed model is useful for understanding the behavioral intention of eco-tourists from different countries and their intention to pay more for ecotourism.

Likewise, Taciano [40] has for years approached studies with the same cross-cultural examination of the environmental attitudes in New Zealand. In their vision, Taciano et al. [41] studied later the same environmental attitudes in different countries, such as Brazil, New Zealand, and South Africa. These results have charted directions for future research that seek to demonstrate that conservation and use are factors distinct from environmental attitudes.

The research also points to the need for an ecological education program about the sustainability of plastic materials based on demonstrations. It is also necessary to build pro-environmental attitudes and behaviors and civic responsibility for the safe and efficient management of plastic waste. Miller et al. [42] examined the relationships between environmental attitudes in 11 countries; the results within countries show that environmental attitudes are a strong predictor, that implies the effectiveness is small in individual countries, highlighting the importance of a more diverse global dimension.

Komatsu et al. [43] carried out a cross-cultural study regarding recycling and the growing positive attitudes for environmental protection and compared single-use plastics from Japan, Canada, and the US. The study highlighted support from older generations, who face difficulties due to new technologies, for disposable plastic. Kaplan Mintz and Kurman [44] investigate at the individual level the effect of recycling in a cross-cultural context between Jewish and Muslim Bedouins. The objective of the study was to investigate the relationships between culture and recycling behavior. Cordano et al. [45] realized a cross-cultural assessment between business students from Chile and the United States regarding pro-environmental behavior; the results present that norms have the strongest relationship with behavioral intention.

Another cross-sectional study was designed by Hien Thi Nguyen et al. [46]. In their study, they realized and analyzed direct and indirect relationships between education, perception, and behaviors towards the problem of plastic waste.

Another cross-sectional model was created also by Kaur Simarjeet and Jeganathan [21] and Shrikrishna Atre [47], who investigate the impact of plastic in our daily life upon human health from another point of view. The researchers [21,47] conducted a study to assess knowledge about the health hazards of plastic use among adolescent girls. The results of the study showed that half of the respondents had poor knowledge; none of them had good knowledge about the health hazards of using plastic.

*2.2. Motivation of the Study*

The plastic waste management problem is a topical issue, but it is perceived differently from one country to another, from individual to individual. The purpose of the present study is to identify the common points and the differences in students' perception and the way to approach and report on this issue regarding plastic waste. The data were used for the development of a common project, FUTURE Bio, and to bring new digital technology to student's education. The new cross-cultural model was created based on a new study having a new target group, namely students from Turkish and Romanian universities, to establish if the model is useful and can investigate the cross-cultural differences of students' knowledge, beliefs, and perception on plastic waste problems.

The objectives of the actual research were:

1.  Identify the strong and weak points of students' information regarding plastic waste.
2.  Identify and measure students' knowledge in plastic waste.
3.  Identify the missing information of students regarding the plastic waste.
4.  Create new tools for students to attract their interest in the field of plastic waste (VR virtual world, digital platform, short videos).
5.  Changing the habits of students regarding the use of plastic.
6.  Acquiring notions about the waste hierarchy.
7.  Knowledge of packaging materials and selective collection.
8.  How to avoid plastic in everyday life.

## 3. Research Methodology

*Materials and Methods*

A total of 589 students were involved in a choice experiment during which a specially designed questionnaire through face-to-face interviews and online was applied between November and December 2022 in two countries with different cultures, amongst students from the North Center University of Baia Mare, a branch of the Technical University of Cluj Napoca, Romania, and Pamukkale University from Denizli and Kirkareli University, both universities from Turkey.

The survey was applied via an online platform using Google Drive forms. The survey had 29 questions, of which 20 questions that tested students' knowledge, beliefs, and preferences in using bioplastic, the recycling process of waste plastic, and environmental pollution, and 5 questions regarding the individual characteristics, then 4 questions regarding students' attitudes.

The survey was structured taking into consideration:

I—Individual characteristics (age, gender, education level, university, faculty, specialization, country).

A—Attitude: participation in conferences, activities organized by university, volunteering activity.

K—Knowledge regarding bioplastic and the impact of plastic on their daily lives.

P—Preferences: if they prefer plastic, or new biodegradable plastic, how they choose the products, if they consider the type of material, if they agree with extra money for new bioplastics.

B—Beliefs: if they have information about plastic waste, plastic pollution, if they believe in replacement of plastic.

The following variables were taken into consideration for the cross-cultural model, as shown in Table 1:

**Table 1.** Survey structure and item for factors taken into consideration.

| Factor | Item | Question |
|---|---|---|
| **Knowledge** | K1 | I know that the air is polluted by burning plastic |
| | K2 | Bioplastics are produced from raw materials that do not harm nature |
| | K3 | Bioplastics are produced from raw materials that do not harm human health |
| | K4 | I know that it is not good to throw plastic products into nature |
| | K5 | I know how to choose products that are less harmful to nature |
| | K6 | I know that I have to throw away the plastic products from the green area |
| | K7 | Bioplastic products do not cause an increase in the greenhouse gas effect |
| **Preferences** | P1 | I prefer to use mesh/cloth/paper bags instead of using disposable bags while shopping |
| | P2 | I prefer to use bioplastic bags for my grocery shopping |
| | P3 | I prefer bioplastic products, even if they are expensive |
| | P4 | I prefer bioplastic products because they degrade earlier in nature |
| | P5 | I prefer the products obtained from the bioplastics industry because they are renewable |
| | P6 | I prefer bioplastic products because they do not harm nature when they decompose |
| | P7 | I prefer bioplastic products as they do not harm human health when degraded |
| | P8 | I prefer to use bioplastic products in the kitchenware |
| | P9 | I prefer not to buy products with nylon additives |
| **Beliefs** | B1 | I believe in biodegradable packages for take-away products |
| | B2 | I believe that at social events biodegradable plastic would be beneficial |
| | B3 | I believe that bioplastic will take the place of conventional polymers |
| | B4 | I believe that pollution studies of plastic should be increased |

To measure students' knowledge and preferences on plastic waste, a similar scale was used by Boca and Saraçli [48] in their research about students' awareness. This scale was used to obtain realistic information. The Likert scale ranged from 1 to 5, where 1—represents "Not at all appropriate" and 5—"Totally appropriate".

The statistical instruments used for data analysis were the SPSS 25 software package and the SMART PLS program. The research between the two countries has been done for a better understanding of students' beliefs regarding a sustainable environment and aimed to examine their preferences on the specific topic of plastic and bioplastic.

This study provides a framework on which the authors build a more in-depth examination of the factors that influence students' perceptions, knowledge, beliefs, and attitudes towards plastic waste, thus a research model, as shown in Figure 1, was used.

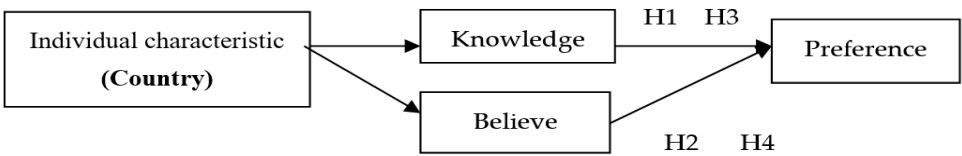

**Figure 1.** Research model for students' cross-cultural model regarding plastic waste.

The hypotheses tested on the perception attitude of students in the present study are:

**H₁.** *Knowledge affects preference regarding attitude of Romanian students towards waste plastic.*

**H₂.** *Belief affects preference regarding attitude of Romanian students towards waste plastic.*

**H₃.** *Knowledge affects preference regarding attitude of Turkish students towards waste plastic.*

**H₄.** *Belief affects preference regarding attitude of Turkish students towards waste plastic.*

As final results of the study will be a cross-cultural model which will help the universities and academic staff to adapt the curricula and academic activities to encourage and to prepare common project and changes of good practice for a better cooperation without borders. Because the cross-cultural model refers to students' culture, the authors take into consideration the country as a cultural factor of influence.

## 4. Results

Cronbach's α coefficient with the value of 0.898 shows that the information obtained from the 589 students from Romania and Turkey from different grades and specialization fields (economics, engineering, medical, textile, electronics) can be taken in consideration.

Using the same survey in Romania and Turkey, at the same time it was possible to establish a cross-cultural model for students from both countries and identify the similarities and the differences between students, taking in consideration specific cultural aspects for each country, traditional style, and different lifestyles. However, the model measures students' levels of perception and knowledge about plastic waste, pollution, and recycling of plastic.

### 4.1. Students Cultural Characteristics

We can observe that, from 589 respondents, 50.08% are Romanian students and 49.92% are Turkish students. For Romania, 59.32% are female and 40.08% are male. For Turkey, 64.63% are female and only 35.37% are male. The students from both countries are from different academic years of study, 44.14% were students in the first year of study and 31.57% were students from the second year, while 24.29% were at master's level and final year.

The results show the students' familiarity with the notion of bioplastics and the information they have obtained over the years through courses or through the medium of the teachers or mass media.

From Table 2, we can observe that, in terms of the field of study in the case of Romanian students, a majority of 38.37% are from the Faculty of Sciences, especially from Economics and Business administration. Of the two countries, 48.56% of the students are from sciences faculties in the case of Turkey; students belonging to the fields of sciences are only 10.19%, from textiles 7.64%, and from medicine 5.43%. A total of 26.65% percent of students are from engineering faculties in Turkey and only 11.71% from Romanian engineering faculties.

**Table 2.** Student distribution according to field of study.

| Question | Specialization | Romania | Turkey | Cumulative Percent |
|---|---|---|---|---|
| | Industrial engineering | 0 | 20 | 3.39 |
| | Faculty of Sciences | 226 | 60 | 48.56 |
| | Denizli Technical Institute | 0 | 59 | 10.02 |
| Which faculty do you study at | Mechatronic Faculty | 0 | 51 | 8.66 |
| | Textile Faculty | 0 | 45 | 7.64 |
| | Faculty of Engineering | 69 | 27 | 16.30 |
| | Medicine | 0 | 32 | 5.43 |
| | Total | 295 | 294 | 100 |

The results are representative for the target group because the students come from different engineering specializations (mechanical, mechatronics, and electrical), textile, or medicine and sciences (management or business administration) who work or have contact with plastic, respectively, bioplastic.

### 4.2. Students' Attitude

Analyzing the data from Table 3, we can observe that 40.41% of the students from both countries participate in and attend conferences about nature protection and waste management, and 59.59% students do not participate in any conferences about plastic waste and nature conservation, which it is a signal that it is necessary to involve students

in different activities, research work, and to encourage them to participate in conferences as team members or to write individual articles to encourage their pioneering spirit.

**Table 3.** Students' participation in activities about plastic waste.

| Question | | Country | | Cumulative Percent |
|---|---|---|---|---|
| | | Romania | Turkey | |
| I participated in conferences on environment and plastic waste | Yes | 82 | 156 | 40.41 |
| | No | 213 | 138 | 59.59 |
| Total | | 295 | 294 | 100 |
| I participated in university activities concerning plastic waste | Yes | 27 | 59 | 14.60 |
| | No | 268 | 235 | 85.40 |
| Total | | 295 | 294 | 100 |

A low percent of 13.92% of Romanian students participate in different conferences and 4.58% participate in activities and events with a focus on environment protection against plastic and plastic pollution. In comparison, there are more Turkish students who are more involved (53.06%) and who show more interest in extra activities.

The biggest value of 81.66% was obtained by Romanian students who never attend conferences, activities, or events on plastic waste and nature conservation, or they do not want to, in comparison with Turkish students, who obtained a low value of 63.33%. So culture (country) does not influence the students' attitudes.

We analyzed the students' distribution between the students' preference for bioplastic and their participation in conferences. The results are shown in Table 4.

**Table 4.** Distribution between students' attendance at conferences and their preference to use bioplastic.

| Question | | Scale | | | | | Cumulative |
|---|---|---|---|---|---|---|---|
| I Prefer Bioplastic Products, Even If They Are Expensive. | | Totally Appropriate | Appropriate | Somewhat Appropriate | Not Appropriate | Not at All Appropriate | Percent % |
| Have you attended a conference on nature conservation before? | Yes | 26 | 63 | 99 | 35 | 15 | 40.41 |
| | No | 45 | 74 | 150 | 44 | 38 | 59.59 |
| Total | | 71 | 137 | 249 | 79 | 53 | 100 |
| Did you take part in environmental activities organized by university? | Yes | 10 | 24 | 35 | 12 | 5 | 14.60 |
| | No | 61 | 113 | 214 | 67 | 48 | 85.40 |
| Total | | 71 | 137 | 249 | 79 | 53 | 100 |

Although the students from both countries prefer bioplastics, a small percentage of 15.11% agree to attend conferences and 20.20% are not interested in the topic, while 42.29% percent of the students are not decided yet regarding the bioplastic subject.

Although universities from both countries organized and hosted environmental activities regarding plastic and bioplastic, only 5.77% of the students participated. So culture does not influence the students' participation in or attendance at different events organized by universities. In conclusion, the students' culture does not affect their attitude As regards the participation and attendance in extracurricular activities, for universities, it is a good signal to improve and to identify new opportunities to attract and involve the new generation.

### 4.3. Students' Knowledge

Students' knowledge regarding plastic products, technological processes, and plastic waste is presented in Table 5.

**Table 5.** Students' distribution in terms of variable knowledge items regarding plastic waste.

| Question | Scale | Country Romania | Turkey | Cumulative Percent |
|---|---|---|---|---|
| Bioplastics are produced from raw materials that do not harm nature. | Totally Appropriate | 69 | 62 | 22.24 |
| | Appropriate | 84 | 147 | 39.22 |
| | Somewhat Appropriate | 62 | 64 | 21.39 |
| | Not Appropriate | 27 | 14 | 6.96 |
| | Not At All Appropriate | 53 | 7 | 10.19 |
| Total | | 295 | 294 | 100 |
| Bioplastics are produced from raw materials that do not harm human health. | Totally Appropriate | 62 | 59 | 20.54 |
| | Appropriate | 82 | 131 | 36.16 |
| | Somewhat Appropriate | 67 | 79 | 24.79 |
| | Not Appropriate | 25 | 17 | 7.13 |
| | Not At All Appropriate | 59 | 8 | 11.38 |
| Total | | 295 | 294 | 100 |
| I know it is important to be informed about the selection of bioplastic products. | Totally Appropriate | 66 | 52 | 20.03 |
| | Appropriate | 72 | 100 | 29.20 |
| | Somewhat Appropriate | 65 | 23 | 14.94 |
| | Not Appropriate | 33 | 109 | 24.10 |
| | Not At All Appropriate | 59 | 10 | 11.70 |
| Total | | 295 | 294 | 100 |
| I know that the air is polluted by burning. | Totally Appropriate | 81 | 90 | 29.03 |
| | Appropriate | 71 | 122 | 32.77 |
| | Somewhat Appropriate | 63 | 63 | 21.39 |
| | Not Appropriate | 41 | 15 | 9.51 |
| | Not At All Appropriate | 39 | 4 | 7.30 |
| Total | | 295 | 294 | 100 |

A total of 61.46% of the students heard or know from mass media that bioplastics are produced from raw materials that do not harm nature and 56.70% know that bioplastics are produced from raw materials that do not harm human health. Small percentages of 19.14% and 18.50% do not have any idea about the topic of plastic waste, maybe because they are not interested in the subject, and they do not care.

Turkish students obtained the highest value, namely 72%, in terms of knowing how to react and inform other people about the plastic problem. A total of 48.81% of the Romanian students know that bioplastic is a smart solution for health protection and the environment, in comparison with Turkish students (64.62%) who recognize the importance of bioplastic. A total of 35.82% of the students do not want to be involved in the subject and take care of nature.

As a conclusion, we can mention that Turkish students are more responsible and more active in environmental activities in comparison with Romanian students. So they have information, they know the importance of pollution, waste management, bioplastic etc., but they do not want to spend extra hours on supplementary activities, or they are not interested in the issue.

*4.4. Students' Preferences*

Table 6 presents the items for students' preferences from both countries which influence their perception of the waste plastic phenomenon. A total of 66.43% of the students prefer biodegradable products and only 20.56% of the students are not decided yet, maybe because the total elimination of plastic takes a long time and patience.

**Table 6.** Students' distribution in terms of items for variable preference using bioplastic.

| Question | Scale | Country Romania | Turkey | Cumulative Percent |
|---|---|---|---|---|
| I prefer the renewable products made from bioplastic. | Totally Appropriate | 95 | 97 | 32.60 |
| | Appropriate | 94 | 129 | 37.86 |
| | Somewhat Appropriate | 51 | 58 | 18.51 |
| | Not Appropriate | 19 | 6 | 4.24 |
| | Not At All Appropriate | 36 | 4 | 6.79 |
| Total | | 295 | 294 | 100 |
| I prefer to use mesh/cloth/paper bags instead of using disposable bags while shopping. | Totally Appropriate | 103 | 105 | 35.31 |
| | Appropriate | 70 | 87 | 26.66 |
| | Somewhat Appropriate | 45 | 76 | 20.54 |
| | Not Appropriate | 33 | 21 | 9.17 |
| | Not At All Appropriate | 44 | 5 | 8.32 |
| Total | | 295 | 294 | 100 |
| I prefer biodegradable plastic. | Totally Appropriate | 97 | 84 | 30.73 |
| | Appropriate | 84 | 140 | 38.03 |
| | Somewhat Appropriate | 55 | 58 | 19.19 |
| | Not Appropriate | 29 | 10 | 6.62 |
| | Not At All Appropriate | 30 | 2 | 5.43 |
| Total | | 295 | 294 | 100 |
| I prefer to use bioplastic bags for my grocery shopping. | Totally Appropriate | 107 | 47 | 26.15 |
| | Appropriate | 65 | 100 | 28.01 |
| | Somewhat Appropriate | 59 | 103 | 27.50 |
| | Not Appropriate | 29 | 31 | 10.18 |
| | Not At All Appropriate | 35 | 13 | 8.15 |
| Total | | 295 | 294 | 100 |

A low percent of 32.08% of Romanian students prefer renewable products in comparison with 38.37% of Turkish students. Also, 58.4% of the students prefer to use bioplastic bags for shopping in an equal percentage between Romanians at 29.20% and Turkish students at 24.96% Here, we have to take into consideration that the new ISO standards obliged all the stores to eliminate plastic bags, so they are following the rules.

Because plastic is around our life, everywhere, even in the kitchen, the students' behavior is similar; 43.6% of them adapt their behavior to the new trend using bamboos, wood tools, and ceramic objects and replace plastic, of which 29.37% are Romanian students and 32.60% are Turkish students. Maybe it is not so difficult for the young generation to adapt and to use the new materials if we take into consideration the new trend in each country, which is returning to the roots, to natural life, and using ceramic and wood objects in our traditional family life, not only as a fashion but as a tradition.

*4.5. Students' Beliefs*

If we look at the variable 'belief of the students', we see that a percentage of 65.56% of the students, regardless of the country, agree that plastic will be replaced in the future (Table 7). A total of 86.2% of students from Turkey and Romania sustain the idea of using biodegradable plastic for social events, of which 61.3% are making efforts to use biodegradable plastic. Students from both countries believe, at 68.76% percent, that disposable biodegradable plastic would be beneficial to use at social events. Turkish students, at 38.71%, totally agree, in comparison with Romanian students, who agree in a lower percentage of 30.06%. Also, Turkish students, at 64.5% percent, present a good behavior and attitude to the environment.

**Table 7.** Students' distribution to items for variable beliefs regarding bioplastic.

| Question | Scale | Country Romania | Turkey | Cumulative Percent |
|---|---|---|---|---|
| I believe in biodegradable packages take away products | Totally Appropriate | 116 | 121 | 40.24 |
| | Appropriate | 67 | 105 | 29.20 |
| | Somewhat Appropriate | 44 | 54 | 16.64 |
| | Not Appropriate | 27 | 10 | 6.28 |
| | Not At All Appropriate | 41 | 4 | 7.64 |
| Total | | 295 | 294 | 100 |
| I believe that at social events (festival, fair, etc.) disposable biodegradable plastic would be beneficial | Totally Appropriate | 110 | 116 | 38.37 |
| | Appropriate | 67 | 112 | 30.39 |
| | Somewhat Appropriate | 58 | 55 | 19.18 |
| | Not Appropriate | 25 | 6 | 5.26 |
| | Not At All Appropriate | 35 | 5 | 5.79 |
| Total | | 295 | 294 | 100 |
| I believe that bioplastic will take the place of conventional polymers | Totally Appropriate | 95 | 91 | 31.58 |
| | Appropriate | 82 | 128 | 35.65 |
| | Somewhat Appropriate | 52 | 59 | 18.85 |
| | Not Appropriate | 20 | 13 | 5.60 |
| | Not At All Appropriate | 46 | 3 | 8.32 |
| Total | | 295 | 294 | 100 |

The results show that students, at 76.45%, believe in research and studies about pollution and plastic replacement and 69.24% percent of students believe that the use of plastics should be generalized in the field of media.

In conclusion, students' beliefs are not influenced by culture, they have similar beliefs regarding the importance of bioplastic. After analysis, the results show that a solution to improve student attitudes and behavior can be found in universities, to involve students in research activities and campaigns dedicated to the environment and waste management.

## 5. Discussion

Using the tree analysis, it was possible to see the connection between the perceptions of students from both countries regarding the use of biodegradable plastic, as shown in Figure 2.

Even though they contribute to the recycling process, price influences their behavior. The students from both countries totally agreed with the increasing price of bioplastic (12.1%) and 78.9% feel that the higher price is somewhat appropriate. A total of 35.5% of Turkish students somewhat agree and 8.8% do not agree. In comparison, Romanian students somewhat agree (16%) with the increasing price and 18.7% consider that it is not an appropriate measure, taking into consideration the social living standard from their country.

Figure 3 completes the cross-cultural model regarding students' participation in different activities organized by universities or society, even though the price of bioplastic is high. Students from both countries (55.5% to be more precise) participate in activities. Even 12.1% totally agree and 23.3% somewhat agree with the increasing price of bioplastic, possibly because they take into consideration that these are the first products using the bioplastic material on market. Those who are on the sidelines and are still undecided about participating in campaigns and other extra activities are at 46.5%. In conclusion, universities as nurseries of future specialists must find solutions and ways to raise awareness and attract students.

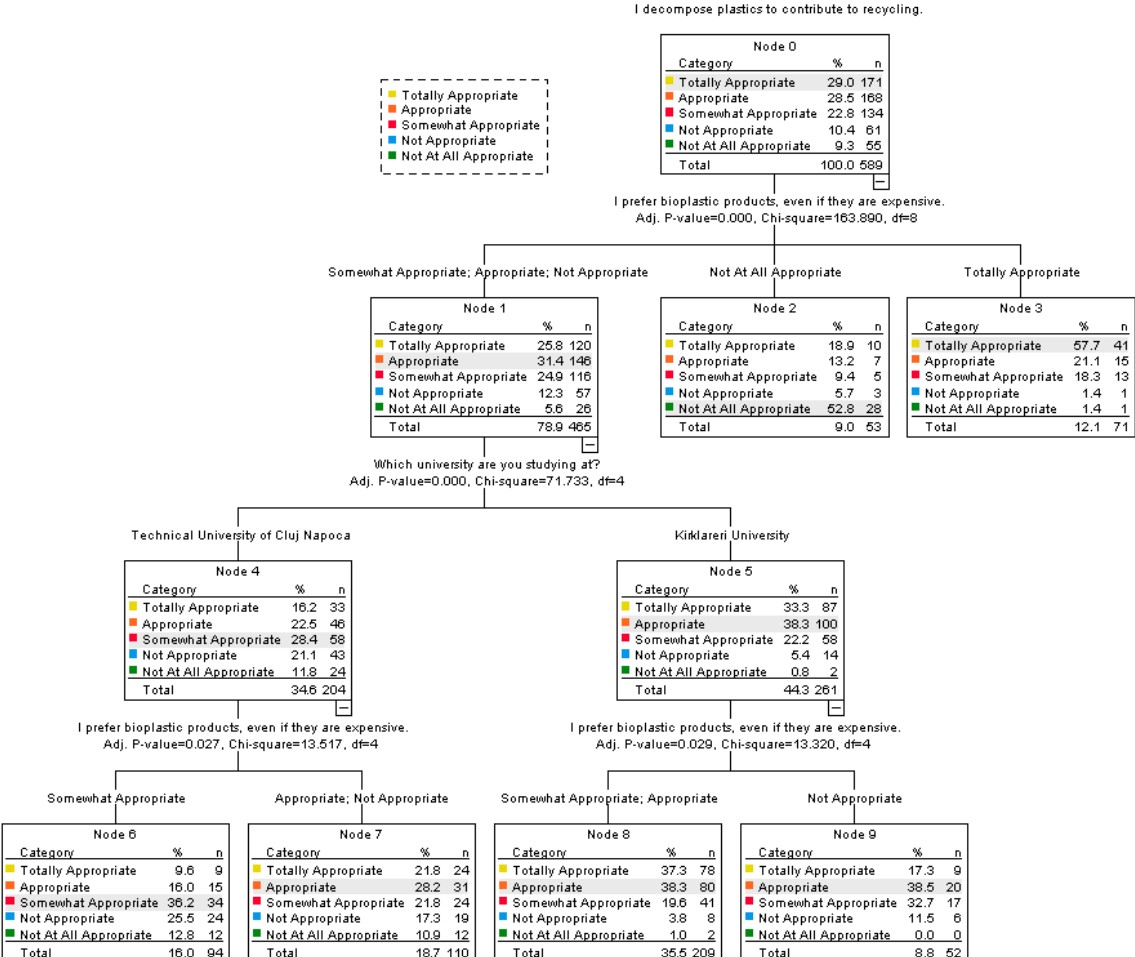

**Figure 2.** Students' preferences and their contribution to the recycling process.

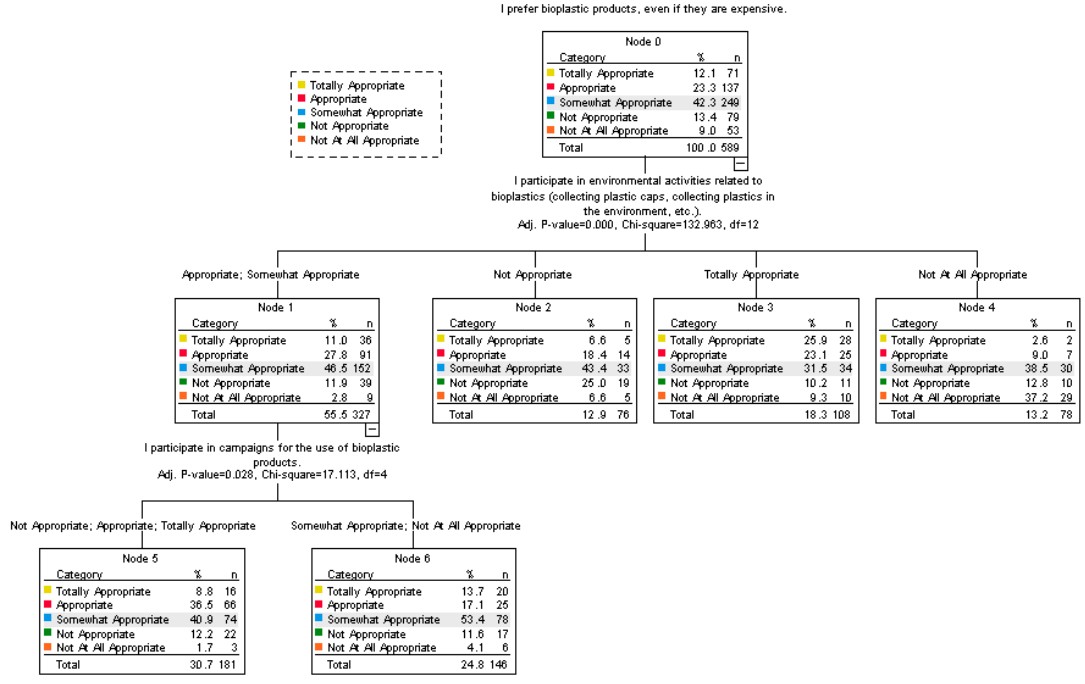

**Figure 3.** Classification results for students' participation in activities and campaigns about bioplastic.

*A Cross-Cultural Model for Students' Perceptions Regarding Plastic Waste*

We took into consideration the database after applying surveys to 589 students from Romanian and Turkish universities. Following the Hair et al. [49], Hair et al. [50], Ringle et al. [51], and Sarstedt [52] research and using the same program, Smart PLS, it was possible to establish the cross model.

We present the SMART PLS program solution, and also the cross-cultural model, taking into consideration the Romanian and Turkish students' knowledge, perceptions, and beliefs and comparing the results. To model the relations among sub-dimensions and to compare Romania and Turkey, we applied PLS-SEM. The results of Factor Loadings, Cronbach's Alpha, CR and AVE values, Fornell–Larcker [53] criterion findings, HTMT criterion findings, parameter estimates, and *t* statistics of the PLS model, and the findings on effect sizes ($f^2$) are given in Tables 8–12 for Romania and Tables 13–17 for Turkey. The structural model for Romania is given in Figure 4 and for Turkey in Figure 5.

**Table 8.** Factor Loadings, Cronbach's Alpha, CR, and AVE values for Romanian students.

| Item/Dimension | Factor Loadings | CA | CR | AVE |
|:---:|:---:|:---:|:---:|:---:|
| | | K | | |
| K1 | 0.663 | | | |
| K2 | 0.820 | | | |
| K3 | 0.752 | | | |
| K4 | 0.743 | 0.809 | 0.860 | 0.473 |
| K5 | 0.707 | | | |
| K6 | 0.490 | | | |
| K7 | 0.584 | | | |
| | | B | | |
| B1 | 0.903 | | | |
| B2 | 0.917 | | | |
| B3 | 0.850 | 0.912 | 0.914 | 0.792 |
| B4 | 0.887 | | | |
| | | P | | |
| P1 | 0.670 | | | |
| P2 | 0.756 | | | |
| P3 | 0.580 | | | |
| P4 | 0.854 | | | |
| P5 | 0.850 | 0.894 | 0.938 | 0.548 |
| P6 | 0.834 | | | |
| P7 | 0.855 | | | |
| P8 | 0.612 | | | |
| P9 | 0.569 | | | |

**Table 9.** Fornell–Larcker criterion findings for Romanian students.

| | B | K | P |
|:---:|:---:|:---:|:---:|
| B | 0.890 | | |
| K | 0.700 | 0.688 | |
| P | 0.844 | 0.809 | 0.740 |

**Table 10.** HTMT criteria findings for Romania.

| | B | K | P |
|:---:|:---:|:---:|:---:|
| B | | | |
| K | 0.797 | | |
| P | 0.895 | 0.962 | |

**Table 11.** Parameter estimates and *t* statistics of the PLS model for Romanian students.

| Hypothesis | Relationship | Parameters ($\beta$) | *t*-Statistics | *p*-Values | Decision |
|---|---|---|---|---|---|
| $H_1$ | B → P | 0.545 | 14.627 | 0.0001 * | Accepted |
| $H_2$ | K → P | 0.425 | 10.165 | 0.0001 * | Accepted |

* $p < 0.01$.

**Table 12.** Findings on effect sizes ($f^2$) for Romania.

| Relationship | $f^2$ Values | Effect Size |
|---|---|---|
| B → P | 0.779 | Strong |
| K → P | 0.478 | Strong |

**Table 13.** Fornell–Larcker criterion findings for Turkish students.

| | B | K | P |
|---|---|---|---|
| B | 0.790 | | |
| K | 0.517 | 0.574 | |
| P | 0.650 | 0.690 | 0.676 |

**Table 14.** HTMT criteria findings for Turkish students.

| | B | K | P |
|---|---|---|---|
| B | | | |
| K | 0.707 | | |
| P | 0.751 | 0.912 | |

**Table 15.** Factor Loadings, Cronbach's Alpha, CR, and AVE values for Turkish students.

| Item/Dimension | Factor Loadings | CA | CR | AVE |
|---|---|---|---|---|
| K | | | | |
| K1 | 0.528 | | | |
| K2 | 0.695 | | | |
| K3 | 0.686 | | | |
| K4 | 0.219 | 0.644 | 0.762 | 0.329 |
| K5 | 0.569 | | | |
| K6 | 0.562 | | | |
| K7 | 0.621 | | | |
| B | | | | |
| B1 | 0.815 | | | |
| B2 | 0.783 | 0.799 | 0.869 | 0.624 |
| B3 | 0.809 | | | |
| B4 | 0.751 | | | |
| P | | | | |
| P1 | 0.562 | | | |
| P2 | 0.526 | | | |
| P3 | 0.488 | | | |
| P4 | 0.810 | | | |
| P5 | 0.774 | 0.845 | 0.879 | 0.457 |
| P6 | 0.831 | | | |
| P7 | 0.823 | | | |
| P8 | 0.591 | | | |
| P9 | 0.561 | | | |

**Table 16.** Parameter estimates and *t* statistics of the PLS model for Turkish students.

| Hypothesis | Relationship | Parameters ($\beta$) | *t*-Statistics | *p*-Values | Decision |
|---|---|---|---|---|---|
| $H_3$ | B → P | 0.400 | 8.279 | 0.0001 * | Accepted |
| $H_4$ | K → P | 0.483 | 10.286 | 0.0001 * | Accepted |

* $p < 0.01$.

**Table 17.** Findings on effect sizes ($f^2$) for Turkish students.

| Relationship | $f^2$ Values | Effect Size |
|---|---|---|
| B → P | 0.289 | Strong |
| K → P | 0.420 | Strong |

There are three criteria to ensure convergent validity in the PLS model:

- first: each standard factor loading of latent variables is greater than 0.5 and statistically significant;
- second: Structural Reliability (CR) and Cronbach's Alpha (CA) values for each structure should be greater than 0.7;
- the third criterion is: the Average Variance Explained (AVE) value should be higher than 0.5 (Fornel and Lacker [53], Hair et al. [54]).

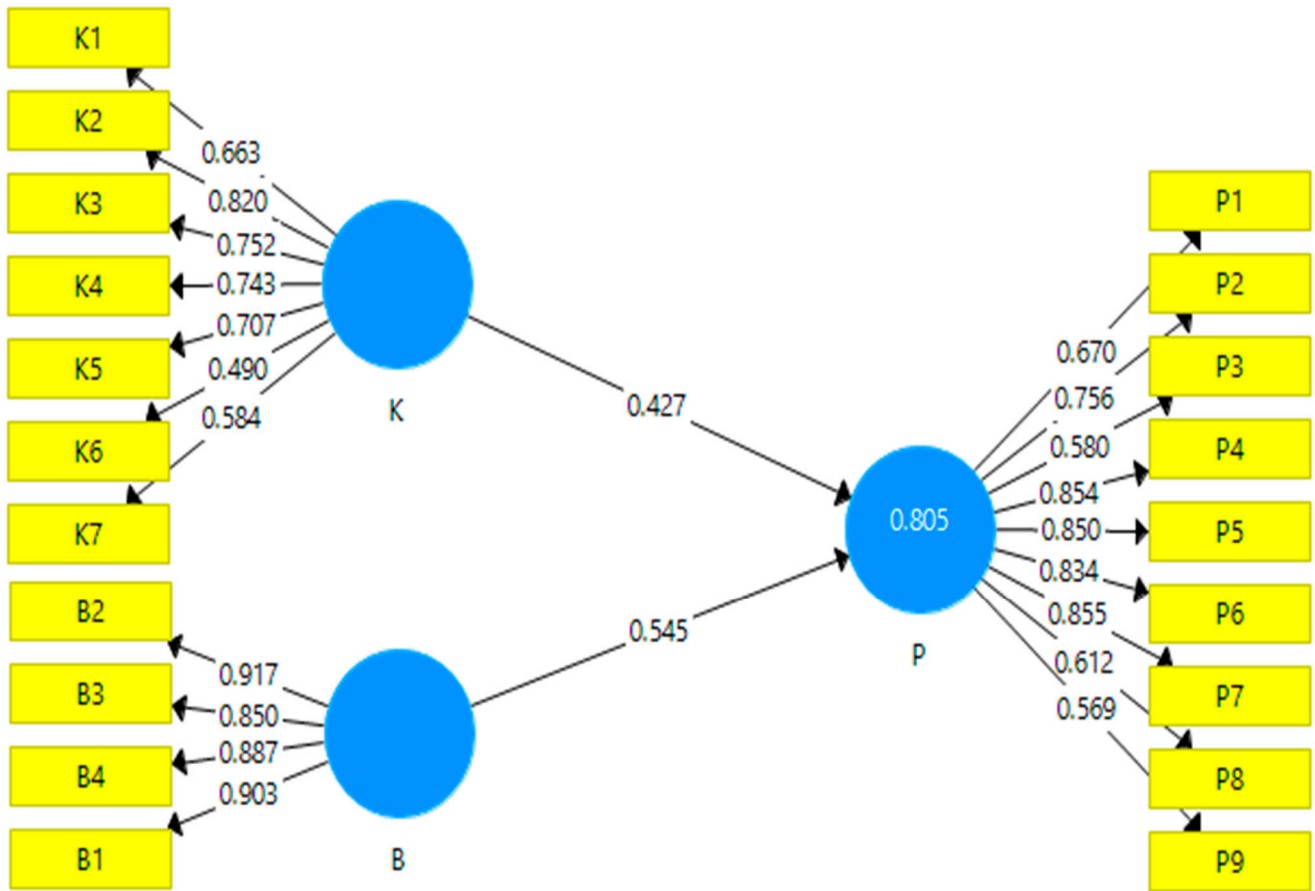

**Figure 4.** PLS path results for Romania.

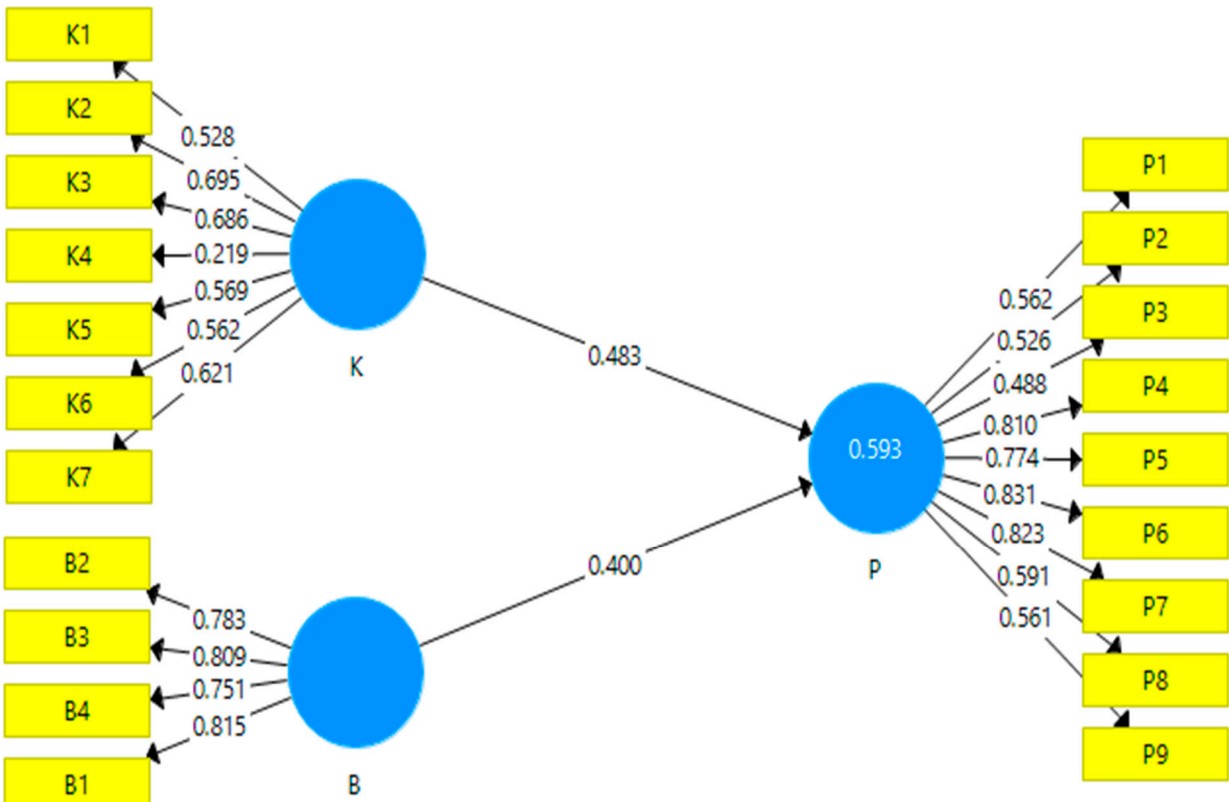

**Figure 5.** PLS path results for Turkish students.

*a.    Cross-cultural model for Romanian students regarding plastic waste*

Factor loadings, CA, CR, AVE values of affective, behavioral, and cognitive dimensions are given in Table 8 for Romanian students. As we can see the latent variable values are valid and accepted. The discriminant validity of the model is checked by comparing the square root of the AVE value for each construct with the correlations between the constructs. Here, it is said that the discriminant validity of the model is ensured if the square root values of the AVE are large [51].

In Table 9, the discriminant validity of the measurement model (Fornell–Larcker criterion) [53] values for all factors are given. When examining the Fornell–Lacker table, the diagonal values represent the square root of the AVE values for each factor, and the off-diagonal values represent the correlation coefficients between the factors.

As an alternative to the Fornell–Lacker criterion, the HTMT criterion represents the geometric mean of factor correlations, which are given in Table 10. In conclusion, the cross model for Romania is available.

The structural model for Romania is given in Figure 4, where we identify the connection and relationship between variables and indicate that knowledge has a statistically significant effect on preferences, with the coefficient of 0.427.

Beliefs have a statistically significant effect on preferences, with the coefficient of 0.545. Knowledge (K) and beliefs (B) explain the 80.5% of preferences (P). Also while:

K2 "Bioplastics are produced from raw materials that do not harm nature" has the highest effect within knowledge, with the coefficient of 0.820,

B1 "I believe in biodegradable packages take away products" has the highest effect within beliefs, with the coefficient of 0.917 and

P7 "I prefer bioplastic products as they do not harm human health when degraded" has the highest effect within preferences, with the coefficient of 0.855.

Also, it can be seen from Table 11, that Hypotheses $H_1$ and $H_2$ for Romania are accepted and path coefficients for the model are statistically significant.

From Table 12, we can observe the effect sizes evaluated as >=0.02 low, >= 0.15 medium, >= 0.35 strong (Hair et al., 1998 [54]). If the VIF value is equal to or greater than five, then it is known that there is a multi-co-linearity problem (Henseler et al. [55]). Since VIF values are less than five, it can be said that there is no multi co-linearity problem. VIF values of this study are between 1.343 and 4.640.

*b.    Cross-cultural model for Turkish students regarding plastic waste*

In Table 13, the discriminant validity of the measurement model (Fornell–Larcker criterion) values for all factors are given again; the discriminant validity of the model is checked by comparing the square root of the AVE value for each construct with the correlations between the constructs. Here, it is said that the discriminant validity of the model is ensured if the square root values of the AVE are large [50].

The alternative to the Fornell–Lacker criterion, for Turkish students for the HTMT criterion are given in Table 14.

Because the proposed model is about cross-culture, we followed similar steps for Turkey. Factor loadings, CA, CR, AVE values of affective, behavioral, and cognitive dimensions are given in Table 15 for Turkish students.

As we can see, the latent variable values are valid and accepted following the standard values mentioned before.

The structural models for Turkey are also given in Figure 5, which indicate that, while the knowledge has a statistically significant effect on preferences with the coefficient of 0.483, beliefs have a statistically significant effect on preferences with the coefficient of 0.400.

Knowledge (K) and beliefs (B) explain 59.3% of preferences (P), similarly to the Romanian model given in Figure 5, while:

K2 "Bioplastics are produced from raw materials that do not harm nature." has the highest effect within knowledge, with the coefficient of 0.695,

B1 "I believe in biodegradable packages take away products." has the highest effect within beliefs, with the coefficient of 0.815 and

P6 "I prefer bioplastic products because they do not harm the nature when they decompose." has the highest effect within preferences, with the coefficient of 0.831.

It can be seen from Table 16 that Hypotheses $H_3$ and $H_4$ for Turkey are accepted and path coefficients for the model are statistically significant.

Since VIF values are all less than five, as with the Romanian results, it can be said that there is no multi-co-linearity problem. VIF values of this study are between 1.052 and 3.497 (Table 17).

In conclusion, results of PLS SEM indicate that both knowledge (K) and beliefs (B) have significant effect on students' preferences (P) for Romania and Turkey. Even some of the criteria for the PLS-SEM model for Turkey are not met, because they are close to critical limits and the evaluation of the same model and comparison of the two countries findings are given in related tables.

*c.    Comparison between the cross-cultural models regarding students' perception of plastic waste*

In the context of students' preference for plastic or bioplastic, their adaptation to new trends on using new material such as bioplastic is present in Table 18, where the maximum values of 0.855 for Romanian students and 0.832 for Turkish students were obtained by item P7—"I prefer bioplastic products as they do not harm human health when degraded" and the lowest value was obtain by the item P2 "I prefer to use bioplastic bags for my grocery shopping for Turkish students and a value of 0.569 for Romanian students for item P9—"I prefer not to buy products with nylon additives prefer biodegradable plastic".

The results show that Turkish and Romanian students are involved in activities specific for the young generation, such as participating in academic life, and they did not pay attention to the small lifestyle details. Also, they do not have money to invest and buy the new products from the market. They have information and knowledge, but they do not

pay too much attention to selection, recycling, and plastic waste, that explains why the variable preference accumulated the lowest values in our cross model.

**Table 18.** Items for students' preferences.

| Item | Question | Romania | Turkey |
|---|---|---|---|
| P1 | I prefer to use mesh/cloth/paper bags instead of using disposable bags while shopping. | 0.670 | 0.562 |
| P2 | I prefer to use bioplastic bags for my grocery shopping. | 0.756 | 0.526 |
| P3 | I prefer bioplastic products, even if they are expensive. | 0.580 | 0.488 |
| P4 | I prefer bioplastic products because they degrade earlier in nature. | 0.854 | 0.810 |
| P5 | I prefer the products obtained from the bioplastics industry because they are renewable. | 0.850 | 0.774 |
| P6 | I prefer bioplastic products because they do not harm nature when they decompose. | 0.834 | 0.831 |
| P7 | I prefer bioplastic products as they do not harm human health when degraded. | 0.855 | 0.832 |
| P8 | I prefer to use bioplastic products in the kitchenware. | 0.612 | 0.591 |
| P9 | I prefer not to buy products with nylon additives. | 0.569 | 0.561 |

For comparison between students' cross-cultural model we chose another variable, namely beliefs, and we selected the following items, as shown in Table 19, and obtained the following results.

**Table 19.** Items for students' beliefs in biodegradable plastic.

| Item | Question | Romania | Turkey |
|---|---|---|---|
| B1 | I believe in biodegradable packages take away products. | 0.917 | 0.783 |
| B2 | I believe that at social events biodegradable plastic would be beneficial. | 0.850 | 0.809 |
| B3 | I believe that bioplastic will take the place of conventional polymers. | 0.887 | 0.751 |
| B4 | I believe that pollution studies of plastic should be increased. | 0.903 | 0.815 |

A maximum value of 0.917 was obtained for item B1—"I believe in biodegradable packages take away products" by Romanian students, which means that students believe that bioplastic represent the future material and will replace plastic as a solution to reduce plastic waste. In exchange, the maximum value of 0.809 was obtained by Turkish students for item B2–"I believe that at social events (festival, fair, etc.) disposable biodegradable plastic would be beneficial" because they know the importance of plastic and the recycling material process, but students from both countries cannot take action, or they do not have the instruments to manage the situation.

For the variable "knowledge", we select the items as in Table 20, the maximum value of 0.820 was obtained by item K2—"Bioplastics are produced from raw materials that do not harm nature" for Romanian students, that means that they have information and knowledge about the importance of the protection of the environment and recycling of plastic.

Also, the same item obtained the maximum value for Turkish students. Along with the activity of recycling, the students have enough information in the field of plastic waste, so they try to adapt to the new trend regarding the selection of non-polluting products.

The lowest value of 0.523 was obtained by Turkish students for item K1—"I know that the air is polluted by burning plastic", and for Romanian students a value of 0.490 for item K6—"I know that I have to throw away the plastic products from the green area" which present a young generation a little bit lazy when it comes to following and observing rules.

The maximum value was obtained by both countries for preference and knowledge variables, which shows us that students are not influenced in their beliefs and culture in our country's case; they are interested in the issue of plastic bringing information without borders regarding the important problems of the planet; they are informed and they know everything there is around the theme. The weak values for variables knowledge, belief, and preferences for students from both countries sustain that students in the future need more knowledge, which can be handled by training, or by universities and organizations which

can involve them in different activities and create an educational culture for the protection of the environment and influences their behavior and attitudes with the new knowledge and information obtained.

**Table 20.** Items for students' knowledge.

| Item | Question | Romania | Turkey |
|------|----------|---------|--------|
| K1 | I know that the air is polluted by burning plastic. | 0.663 | 0.528 |
| K2 | Bioplastics are produced from raw materials that do not harm nature. | 0.820 | 0.695 |
| K3 | Bioplastics are produced from raw materials that do not harm human health. | 0.752 | 0.686 |
| K4 | I know that it is not good to throw plastic products into nature. | 0.743 | 0.219 |
| K5 | I know how to choose products that are less harmful to nature. | 0.707 | 0.569 |
| K6 | I know that I have to throw away the plastic products from the green area. | 0.490 | 0.562 |
| K7 | Bioplastic products do not cause an increase in the greenhouse gas effect. | 0.584 | 0.621 |

## 6. Limitations of the Work

This study has some limitations. First of all, the construction of the model of variables taken into consideration in the evaluation of knowledge about plastic waste is not comprehensive enough. Currently, there is no unified system of indicators, and different indicators can lead to different results; therefore, the results here should be further objectively validated. Second, plastic waste is influenced by many factors, and the dominant factors vary. Third, due to data availability limitations, the duration of this study is relatively short. Fourth, more comprehensive research is needed, starting from the characteristics of country, city university, researchers should select cities, universities with similar development characteristics for comparative studies. Because of the respondents' cultural backgrounds or perspectives on certain phenomena, this could affect the legitimacy of the study in using the model. It should be noted that the research problem has been stated and the data collection process has been carried out properly. Future research can use our study as a starting point to track changes over time, expand the study area, and further examine academic staff or industry employees.

The cross-cultural model was applied only to students from three universities, from two countries, Romania and Turkey. It is useful for universities to identify the students' level of knowledge regarding different topics and to obtain solutions to increase students' interest. Also it would be interesting to apply the same model amongst academic staff and workers from industry who are involved in the technological process and plastic waste management.

From the statistical point of view, there are other methods to determine the statistically significant variables for the categorical dependent variable, to determine the effective factors on students preferences about bioplastic, and to give the results more visually; in this study we applied CHAID (Chi-Squared Automatic Interaction Detection) analysis. On the other hand the relationships among the dimensions can be modeled by either covariance-based SEM (CB-SEM) or partial least squares SEM (PLS-SEM). Because PLS-SEM is primarily used to develop theories in exploratory research, in this study, we preferred this approach. Sample size of this study (295 for Romania, 294 for Turkey, and 589 in total) is large enough to apply SEM at an acceptable power (min. 80%) and all the statistical analysis are concluded at 95% confidence level.

## 7. Conclusions

The proposed model starts from cultural differences (country), hypothetically identified as an impediment, and was transformed into a successful instrument by turning obstacles into opportunities through a frontal approach.

The challenge of today due to globalization sustains the importance of understanding other cultures and the importance of culture in production, business, and quality management, and encourages education for new challenges.

The study proposed a cross-cultural model regarding students' perception from Romania and Turkey and identified students' preferences regarding plastic waste.

The results confirm that country, or traditional national lifestyle, cannot influence the students' preferences or knowledge in the digital era when information is without borders. The young generation is very curious and obtains everything using digital technology but they do not pay attention to the small details in their daily lives, maybe because they are concentrated on their education.

The cross-model wants to highlight the preferences and knowledge of students about plastic waste and the results showed that there is no differences between students' information and students' knowledge about plastic waste between the two countries; also gender or specialization have no influence on the perception of bioplastic.

Another common point for students from both countries is that, at a low percent, they participate in and attend conferences about nature protection, plastic waste, and a constant percentage of students do not participate in any conferences about nature conservation or recycling materials and they are on standby, maybe because it is not specific to their field of study.

As a conclusion, we can mention that Turkish students are more responsible and more active in environmental activities regarding plastic waste in comparison with Romanian students. Turkish students are more careful with recycling waste plastic, and they choose products that are less harmful to nature in comparison with Romanian students.

As a final conclusion university can apply the 7Ds circle, as shown in Figure 6:

1.  direction to give students information regarding the plastic waste using digital technology (digital platform for courses and lectures), small videos to understand better the phenomenon;
2.  decision to involve students in laboratory activities and research work as a practical period;
3.  documentation—working together in same research topic using good practice and change of methods and results, transfer of information between universities;
4.  diversification—harmonization with the last trends from the industry and the market, create and develop new specialization, new jobs on the market because of plastic waste;
5.  development—harmonization with the new digital technology (mobile phone, digital platforms, virtual reality, ZOOM meeting); universities can adapt the lectures and curricula in correlation with the students' needs, share ideas and information;
6.  desire—the needs of the young generation to adapt to the new trends;
7.  durability—the open access to instruments used in and by universities can attract the young generation in research or volunteer activities.

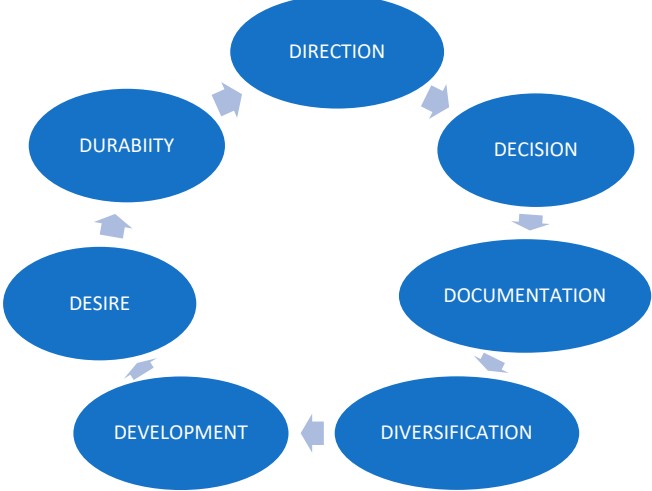

**Figure 6.** The 7Ds for cross-cultural development.



**Author Contributions:** Conceptualization, G.D.B., A.I., S.S. and E.Ç.; methodology, G.D.B., A.I., S.S. and E.Ç.; validation; G.D.B., A.I., S.S. and E.Ç.; writing—original draft preparation, G.D.B., A.I., S.S. and E.Ç.; writing—review and editing. All authors have read and agreed to the published version of the manuscript.

**Funding:** This research was funded by Erasmus Plus project, grant number TR01-KA220 HED-000032160 FUTURE Bio.

**Institutional Review Board Statement:** The study was conducted in accordance with the Declaration of Helsinki, and approved by the Ethics Committee of Pamukkale University (protocol code E-93803232-622.02114625/18-14/07.10.2021) and Ethics Committee of Technical University of Cluj Napoca (CEU 515/20/03/2023).

**Informed Consent Statement:** Not applicable.

**Data Availability Statement:** Data are unavailable due to privacy and ethical restrictions.

**Conflicts of Interest:** The authors declare no conflict of interest.

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
