# Peer review of "A Cross-Cultural Analysis for Plastic Waste Perception of Students from Romania and Turkey"

_sustainability, doi:10.3390/su152416594_

Round 1

Reviewer 1 Report

Comments and Suggestions for Authors

The manuscript is very difficult to understand. At its present state, I think it cannot be published.

Although they have cited many related studies, the research gap needs to be positioned better. What unique knowledge are they trying to contribute to the pool of existing ones, which they have cited to be already very abundant. For instance, they may need to emphasise better what are they are trying to focus on in this study. Are they emphasising how the difference of culture impact plastic recycling behaviours? What cultural characteristics are relevant? They might wish to introduce unique cultural features of Turkey and Romania that could be relevant to plastic recycling.  Also, why survey university students? It might be beneficial to justify their reason in the paper.

I think their sample is big enough; however, they did not specify how the sampling was decided. In the discussion of the cross model results, it is not clear what are the implications of the differences between Romanian and Turkish students. It seems they lumped the samples together in the analysis of the cross model. If the paper is comparative, it seems they did not do any comparison with this model for the two subgroups.

Although they are lifting the model from a previous study, it would have been useful for the readers that they also explain a bit the cross model in this paper.

Comments on the Quality of English Language

As I mentioned earlier, the manuscript is very difficult to understand. Many sentences/phrases are not clear:

For instance, what does "can no longer be considered an effort" [36] mean? What do you mean "there are limits to plastic" [48]. Also lines 354-355, 417-421, 428-430…  These are just examples, and there are many more. The abstract also needs to be improved.

Author Response

Response to Reviewer 1 Comments

Dear Reviewer

First of all we want to thank you for your attention and time to review our articles and for your suggestions

The manuscript is very difficult to understand. At its present state, I think it cannot be published.

Although they have cited many related studies, the research gap needs to be positioned better.

Point 1

 What unique knowledge are they trying to contribute to the pool of existing ones, which they have cited to be already very abundant.

Response 1

Motivation of the Study

The plastic waste management problem is a topical issue, but it is perceived differently from one country to another, from individual to individual. The purpose of the present study is to identify the common points and the differences in students’ perception and the way to approach and report on this issue regarding plastic waste. The data was used for the development of a common project, FUTURE Bio, and to bring new digital technology in student’s education. The new cross-cultural model was created based on a new study having a new target group, namely students’ from Turkish and Romanian universities, and establish if the model is useful and can investigate the cross-cultural differences of students’ knowledge, believes and perception on plastic waste problems.

The objectives of the actual research were:

  1. Identify the strong and weak points of students’ information regarding plastic waste;
  2. Identify and measure students’ knowledge in plastic waste;
  3. Identify the missing information of students regarding the plastic waste;
  4. Create new tools for students to attract their interest in the field of plastic waste (VR virtual world, digital platform, short videos);
  5. Changing the habits of students regarding the use of plastic;
  6. Acquiring notions about the waste hierarchy;
  7. Knowledge of packaging materials and selective collection;
  8. How to avoid plastic in everyday life

Point 2

For instance, they may need to emphasise better what are they are trying to focus on in this study.

Response2

The survey was applied on an online platform using Google Drive forms. The survey had 29 questions of which 20 questions that tested students' knowledge, believes and preferences in using bioplastic, the recycling process of waste plastic and environmental pollution, and 5 questions regarding the individual characteristics, then 4 questions regarding students’ attitude.

The survey was structure taking in consideration:

I—Individual characteristics (age, gender, education level, university, faculty, specialization, country);

A—Attitude: participation to conferences, activities organized by university, volunteering activity.

K—Knowledge regarding bioplastic and the impact of plastic in their daily life;

P —Preferences: if they prefer plastic, or new biodegradable plastic, how they choose the products, if they consider the type the material, if they agree with extra money for new bioplastic;

B – Believes: if they have information about plastic waste, plastic pollution, if they believe in replacement of plastic.

The following variables were taken in consideration for the cross-cultural model, as shown in Table 1:

Table 1.  Survey structure and item for factors taken in consideration

Factor

KNOWLEDGE

Item  

Question

K1

I know that the air is polluted by burning plastic

K2

Bioplastics are produced from raw materials that do not harm nature

K3

Bioplastics are produced from raw materials that do not harm human health

K4

I know that it is not good to throw plastic products into nature

K5

I know how to choose products that are less harmful to nature

K6

I know that I have to throw away the plastic products from the green area

K7

Bioplastic products do not cause an increase in the greenhouse gas effect

PREFERENCE

P1

I prefer to use mesh/cloth/paper bags instead of using disposable bags while shopping

P2

I prefer to use bioplastic bags for my grocery shopping.

P3

I prefer bioplastic products, even if they are expensive

P4

I prefer bioplastic products because they degrade earlier in nature.

P5

I prefer the products obtained from the bioplastics industry because they are renewable

P6

I prefer bioplastic products because they do not harm nature when they decompose

P7

I prefer bioplastic products as they do not harm human health when degraded

P8

I prefer to use bioplastic products in the kitchenware

P9

I prefer not to buy products with nylon additives

BELIEVES

B1

I believe in biodegradable packages for take-away products

B2

I believe that at social events biodegradable plastic would be beneficial

B3

I believe that bio plastic will take the place of conventional polymers

B4

I believe that pollution studies of plastic should be increased

To measure students’ knowledge and preferences on plastic waste, a similar scale used Boca and Saraçli [48] in their research about students’ awareness was used to obtain realistic information. The Likert scale ranged from 1 to 5, where 1- represent “Not at all appropriate” and 5 – “Totally appropriate”.

The statistical instruments used for data analysis was the SPSS 25 software package and the SMART PLS4 program. The research between the two countries has been done for a better understanding of students’ beliefs regarding sustainable environment and aimed to examine their preferences on the specific topic of plastic and bioplastic.

This study provides a framework on which the authors build a more in-depth examination of the factors that influence students’ perception, knowledge, beliefs and attitude towards plastic waste, thus a research model as shown in Figure 1 was used.

Figure 1. Research model for Students’ Cross Cultural Model regarding plastic waste

The hypotheses tested on the perception attitude of students in the present study are:

H1: Knowledge effects Preference regarding attitude of Romanian students towards waste plastic;

H2: Believes effects Preference regarding attitude of Romanian students towards waste plastic;

H3: Knowledge effects Preference regarding attitude of Turkish students towards waste plastic;

H4: Believes effects Preference regarding attitude of Turkish students towards waste plastic.

As final results of the study will be a cross-cultural model which will help the universities and academic staff to adapt the curricula and academic activities to encourage and to prepare common project and changes of good practice for a better cooperation without borders. Because the cross-cultural model refers to students’ culture, the authors take in consideration the country as a cultural factor of influence.

Point 3

 Are they emphasising how the difference of culture impact plastic recycling behaviours?

Response 3

We create the separate model for Romanian students and separate same model for Turkish students and then compare the results.

The structural model for Romania is given in Figure 4 where we identify the connection and relationship between variables and indicate that while the Knowledge has a statistically significant effect on Preferences with the coefficient of 0.427.

Figure 4. PLS Path Results for Romania

Beliefs have a statistically significant effect on Preferences with the coefficient of 0.545. Knowledge (K) and Beliefs (B) explain the 80.5% of Preferences (P). Also while:

K2 “Bioplastics are produced from raw materials that do not harm nature.’’ has the highest effect within Knowledge with the coefficient of 0.820,

B1 “I believe in biodegradable packages take away products’’ has the highest effect within Beliefs with the coefficient of 0.917 and

P7 “I prefer bio plastic products as they do not harm human health when degraded’’ has the highest effect within Preferences with the coefficient of 0.855.

Also it can be seen from Table 12, that Hypothesis H1 and H2 for Romania are accepted and path coefficients for the model are statistically significant.

Table 12. Parameter estimates and t statistics of the PLS model for Romanian students ‘

Hypothesis

Relationship

Parameters (β)

t- statistics

p- values

Decision

H1

B  P

0.545

14.627

0.0001*

Accepted

H2

K  P

0.425

10.165

0.0001*

Accepted

*p<0.01

From Table 13 we can observe the effect sizes evaluated as >=0.02 low, >= 0.15 medium, >= 0.35 strong (Hair et al., 1998). If the VIF value is equal to or greater than 5, then it is known that there is a multi-co-linearity problem (Henseler et al.[55]). Since VIF values ​​are less than 5, it can be said that there is no multi co-linearity problem. VIF values of this study are between 1.343 and 4.640.

                Table 13. Findings on effect sizes (f 2) for Romania.

Relationship

f 2 Values

Effect Size

B  P

0.779

Strong

K  P

0.478

Strong

  1. Cross cultural model for Turkish students regarding plastic waste

In Table 14, the discriminant validity of the measurement model (Fornell-Larcker criterion) values for all factors are given again, the discriminant validity of the model is checked by comparing the square root of the AVE value for each construct with the correlations between the constructs. Here, it is said that the discriminant validity of the model is ensured if the square root values of the AVE are large [50].

                        Table 14. Fornell-Larcker criterion findings for Turkish students’

B

K

P

B

0,790

K

0,517

0,574

P

0,650

0,690

0,676

The alternative to the Fornell-Lacker criterion, for Turkish students for the HTMT criterion are given in Table 15.

                                Table 15. HTMT criteria findings for Turkish students’

B

K

P

B

K

0,707

P

0,751

0,912

Because the proposed model is about cross-culture, we followed similar steps for Turkey. Factor loadings, CA, CR, AVE values of affective, behavioral and cognitive dimensions are given in Table 16 for Turkish students.

                                    Table 16. Factor Loadings, Cronbach's Alpha, CR and AVE values for Turkish students

Item/Dimension

Factor Loadings

CR

AVE

K

K1

0,528

0,644

0,762

0,329

K2

0,695

K3

0,686

K4

0,219

K5

0,569

K6

0,562

K7

0,621

B

B1

0,815

0,799

0,869

0,624

B2

0,783

B3

0,809

B4

0,751

P

P1

0,562

0,845

0,879

0,457

P2

0,526

P3

0,488

P4

0,810

P5

0,774

P6

0,831

P7

0,823

P8

0,591

P9

0,561

As we can see the latent variables values are valid and accepted following the standard values mentioned before.

The structural models for Turkey are also given in Figure 5 which indicate that while the Knowledge has a statistically significant effect on Preferences with the coefficient of 0.483,

Beliefs has a statistically significant effect on Preferences with the coefficient of 0.400.

Knowledge (K) and Beliefs (B) explain the 59.3% of Preferences (P) similarly with the Romanian model given in Figure 5, while:

K2 “Bioplastics are produced from raw materials that do not harm nature.’’ has the highest effect within Knowledge with the coefficient of 0.695,

B1 “I believe in biodegradable packages take away products.’’ has the highest effect within Beliefs with the coefficient of 0.815 and

P6 “I prefer bioplastic products because they do not harm the nature when they decompose.’’ has the highest effect within Preferences with the coefficient of 0.831.

Figure 5. PLS Path Results for Turkish students’

It can be seen from Table 17 that Hypothesis H3 and H4 for Turkey are accepted and path coefficients for the model are statistically significant.

Table 17. Parameter estimates and t statistics of the PLS model for Turkish students

Hypothesis

Relationship

Parameters (β)

t- statistics

p- values

Decision

H3

B  P

0.400

8.279

0.0001*

Accepted

H4

K  P

0.483

10.286

0.0001*

Accepted

*p<0.01

Since VIF values ​​are all less than 5 as Romanian results, it can be said that there is no multi-co-linearity problem. VIF values of this study are between 1.052 and 3.497 (Table 18).

                                 Table 18. Findings on effect sizes (f 2) for Turkish students

Relationship

f 2 Values

Effect Size

B  P

0.289

Strong

K  P

0.420

Strong

In conclusion results of PLS SEM indicate that both Knowledge (K) and Beliefs (B) have significant effect on students Preferences (P) for Romania and Turkey. Even some of the criteria for PLS-SEM model for Turkey do not meet, because they are close to critical limits and to evaluate the same model and compare two countries findings are given in related tables.

Point 4

 What cultural characteristics are relevant?

Response 4

This study provides a framework on which the authors build a more in-depth examination of the factors that influence students’ perception, knowledge, beliefs and attitude towards plastic waste, thus a research model as shown in Figure 1 was used.

Figure 1. Research model for Students’ Cross Cultural Model regarding plastic waste

The hypotheses tested on the perception attitude of students in the present study are:

H1: Knowledge effects Preference regarding attitude of Romanian students towards waste plastic;

H2: Believes effects Preference regarding attitude of Romanian students towards waste plastic;

H3: Knowledge effects Preference regarding attitude of Turkish students towards waste plastic;

H4: Believes effects Preference regarding attitude of Turkish students towards waste plastic.

Point 5

 They might wish to introduce unique cultural features of Turkey and Romania that could be relevant to plastic recycling. 

Response 5

Point 6

 Also, why survey university students? It might be beneficial to justify their reason in the paper.

Response 6

Point 7

I think their sample is big enough; however, they did not specify how the sampling was decided.

Response 7

Point 8

 In the discussion of the cross model results, it is not clear what are the implications of the differences between Romanian and Turkish students.

Response 8

Point 9

It seems they lumped the samples together in the analysis of the cross model.

Response 9

Point 10

 If the paper is comparative, it seems they did not do any comparison with this model for the two subgroups

Response 10

. 5.3. A Cross Cultural Model for Students Perception Regarding Plastic Waste

            We took into consideration the database after applying surveys to 589 students from Romanian and Turkish universities. Following Hair et al. [49], Hair et al. [50], Ringle et al.[51] and Sarstedt [52] researches and using the same program Smart PLS 4 was possible to establish the cross model.            

            We present the SMART PLS 4 program solution, and also the cross-cultural model taking into consideration the Romanian and Turkish students’ knowledge, perception and students’ beliefs and comparing the results. To model the relations among sub-dimensions and compare Romania and Turkey we applied PLS-SEM. The results of Factor Loadings, Cronbach's Alpha, CR and AVE values, Fornell-Larcker [53] criterion findings, HTMT criteria findings, parameter estimates and t statistics of the PLS model, and findings on effect sizes (f 2) are given in Tables 9-13 for Romania and Tables 14-18 for Turkey. The structural model for Romania is given in Figure 4 and for Turkey in Figure 5.

There are three criteria to ensure convergent validity in the PLS model

  • first: each standard factor loading of latent variables is greater than 0.5 and statistically significant;
  • second: Structural Reliability (CR) and Cronbach's Alpha (CA) values for each structure should be greater than 0.7;
  • the third criterion is: The Average Variance Explained (AVE) value should be higher than 0.5 (Fornel & Lacker [53], Hair et al.[54]).

  1. Cross cultural model for Romanian students regarding plastic waste

Factor loadings, CA, CR, AVE values of affective, behavioral and cognitive dimensions are given in Table 9 for Romanian students. As we can see the latent variables values are valid and accepted. The discriminant validity of the model is checked by comparing the square root of the AVE value for each construct with the correlations between the constructs. Here, it is said that the discriminant validity of the model is ensured if the square root values of the AVE are large [51].

                         Table 9. Factor Loadings, Cronbach's Alpha, CR and AVE values for Romanian students’

Item/Dimension

Factor Loadings

CR

AVE

K

K1

0.663

0.809

0.860

0.473

K2

0.820

K3

0.752

K4

0.743

K5

0.707

K6

0.490

K7

0.584

B

B1

0.903

0.912

0.914

0.792

B2

0.917

B3

0.850

B4

0.887

P

P1

0.670

0.894

0.938

0.548

P2

0.756

P3

0.580

P4

0.854

P5

0.850

P6

0.834

P7

0.855

P8

0.612

P9

 0.569

In Table 10, the discriminant validity of the measurement model (Fornell-Larcker criterion) [53] values for all factors are given. When examining the Fornell-Lacker table, the diagonal values ​​represent the square root of the AVE values ​​for each factor, and the off-diagonal values ​​represent the correlation coefficients between the factors.

                              Table 10. Fornell-Larcker criterion findings for Romanian students’

B

K

P

B

0,890

K

0,700

0,688

 P

 0,844

 0,809

0,740

As an alternative to the Fornell-Lacker criterion, the HTMT criterion represents the geometric mean of factor correlations which are given in Table 11. In conclusion the cross model for Romania is available.

                               Table 11. HTMT criteria findings for Romania.

B

K

P

B

K

0,797

P

0,895

0,962

The structural model for Romania is given in Figure 4 where we identify the connection and relationship between variables and indicate that while the Knowledge has a statistically significant effect on Preferences with the coefficient of 0.427.

Figure 4. PLS Path Results for Romania

Beliefs have a statistically significant effect on Preferences with the coefficient of 0.545. Knowledge (K) and Beliefs (B) explain the 80.5% of Preferences (P). Also while:

K2 “Bioplastics are produced from raw materials that do not harm nature.’’ has the highest effect within Knowledge with the coefficient of 0.820,

B1 “I believe in biodegradable packages take away products’’ has the highest effect within Beliefs with the coefficient of 0.917 and

P7 “I prefer bio plastic products as they do not harm human health when degraded’’ has the highest effect within Preferences with the coefficient of 0.855.

Also it can be seen from Table 12, that Hypothesis H1 and H2 for Romania are accepted and path coefficients for the model are statistically significant.

Table 12. Parameter estimates and t statistics of the PLS model for Romanian students ‘

Hypothesis

Relationship

Parameters (β)

t- statistics

p- values

Decision

H1

B  P

0.545

14.627

0.0001*

Accepted

H2

K  P

0.425

10.165

0.0001*

Accepted

*p<0.01

From Table 13 we can observe the effect sizes evaluated as >=0.02 low, >= 0.15 medium, >= 0.35 strong (Hair et al., 1998). If the VIF value is equal to or greater than 5, then it is known that there is a multi-co-linearity problem (Henseler et al.[55]). Since VIF values ​​are less than 5, it can be said that there is no multi co-linearity problem. VIF values of this study are between 1.343 and 4.640.

                Table 13. Findings on effect sizes (f 2) for Romania.

Relationship

f 2 Values

Effect Size

B  P

0.779

Strong

K  P

0.478

Strong

  1. Cross cultural model for Turkish students regarding plastic waste

In Table 14, the discriminant validity of the measurement model (Fornell-Larcker criterion) values for all factors are given again, the discriminant validity of the model is checked by comparing the square root of the AVE value for each construct with the correlations between the constructs. Here, it is said that the discriminant validity of the model is ensured if the square root values of the AVE are large [50].

                        Table 14. Fornell-Larcker criterion findings for Turkish students’

B

K

P

B

0,790

K

0,517

0,574

P

0,650

0,690

0,676

The alternative to the Fornell-Lacker criterion, for Turkish students for the HTMT criterion are given in Table 15.

                                Table 15. HTMT criteria findings for Turkish students’

B

K

P

B

K

0,707

P

0,751

0,912

Because the proposed model is about cross-culture, we followed similar steps for Turkey. Factor loadings, CA, CR, AVE values of affective, behavioral and cognitive dimensions are given in Table 16 for Turkish students.

                                    Table 16. Factor Loadings, Cronbach's Alpha, CR and AVE values for Turkish students

Item/Dimension

Factor Loadings

CR

AVE

K

K1

0,528

0,644

0,762

0,329

K2

0,695

K3

0,686

K4

0,219

K5

0,569

K6

0,562

K7

0,621

B

B1

0,815

0,799

0,869

0,624

B2

0,783

B3

0,809

B4

0,751

P

P1

0,562

0,845

0,879

0,457

P2

0,526

P3

0,488

P4

0,810

P5

0,774

P6

0,831

P7

0,823

P8

0,591

P9

0,561

As we can see the latent variables values are valid and accepted following the standard values mentioned before.

The structural models for Turkey are also given in Figure 5 which indicate that while the Knowledge has a statistically significant effect on Preferences with the coefficient of 0.483,

Beliefs has a statistically significant effect on Preferences with the coefficient of 0.400.

Knowledge (K) and Beliefs (B) explain the 59.3% of Preferences (P) similarly with the Romanian model given in Figure 5, while:

K2 “Bioplastics are produced from raw materials that do not harm nature.’’ has the highest effect within Knowledge with the coefficient of 0.695,

B1 “I believe in biodegradable packages take away products.’’ has the highest effect within Beliefs with the coefficient of 0.815 and

P6 “I prefer bioplastic products because they do not harm the nature when they decompose.’’ has the highest effect within Preferences with the coefficient of 0.831.

Figure 5. PLS Path Results for Turkish students’

It can be seen from Table 17 that Hypothesis H3 and H4 for Turkey are accepted and path coefficients for the model are statistically significant.

Table 17. Parameter estimates and t statistics of the PLS model for Turkish students

Hypothesis

Relationship

Parameters (β)

t- statistics

p- values

Decision

H3

B  P

0.400

8.279

0.0001*

Accepted

H4

K  P

0.483

10.286

0.0001*

Accepted

*p<0.01

Since VIF values ​​are all less than 5 as Romanian results, it can be said that there is no multi-co-linearity problem. VIF values of this study are between 1.052 and 3.497 (Table 18).

                                 Table 18. Findings on effect sizes (f 2) for Turkish students

Relationship

f 2 Values

Effect Size

B  P

0.289

Strong

K  P

0.420

Strong

In conclusion results of PLS SEM indicate that both Knowledge (K) and Beliefs (B) have significant effect on students Preferences (P) for Romania and Turkey. Even some of the criteria for PLS-SEM model for Turkey do not meet, because they are close to critical limits and to evaluate the same model and compare two countries findings are given in related tables.

  1. Comparison between the cross-cultural models regarding students’ perception of plastic waste

 In the context of students’ preference for plastic or bioplastic, their adaptation to new trends on using new material such as bioplastic is present in Table 19 where the maximum values of 0.855 for Romanians students and 0.832 for Turkish students was obtained by item P7- “I prefer bio plastic products as they do not harm human health when degraded” and the lowest value was obtain by the item P2 “I prefer to use bio plastic bags for my grocery shopping for Turkish students and a value of 0.569 for Romanian students for item P9- “I prefer not to buy products with nylon additives prefer biodegradable plastic’’.

               Table 19. Items for students’ preferences

Item

Question

Romania

Turkey

P1

I prefer to use mesh/cloth/paper bags instead of using disposable bags while shopping.

0.670

0.562

P2

I prefer to use bioplastic bags for my grocery shopping.

0.756

0.526

P3

I prefer bioplastic products, even if they are expensive

0.580

0.488

P4

I prefer bioplastic products because they degrade earlier in nature.

0.854

0.810

P5

I prefer the products obtained from the bioplastics industry because they are renewable.

0.850

0.774

P6

I prefer bioplastic products because they do not harm nature when they decompose.

0.834

0.831

P7

I prefer bioplastic products as they do not harm human health when degraded.

0.855

0.832

P8

I prefer to use bioplastic products in the kitchenware.

0.612

0.591

P9

I prefer not to buy products with nylon additives

0.569

0.561

The results show that Turkish and Romanian students are involved in activities specific for the young generation, such as participating in the academic life and they did not pay attention to the small lifestyle details. Also they do not have money to invest and buy the new products from the market. They have information and knowledge but they do not pay too much attention to selection, recycling and plastic waste, that explains why the variable preference accumulate lowest values in our cross model.

For comparison between students’ cross cultural model we chose another variable, namely beliefs, and we selected the following items as shown in Table 20 and obtained the following results.

Table 20. Items for Students’ belief in biodegradable plastic.

Item

Question

Romania

Turkey

B1

I believe in biodegradable packages take away products.

0.917

0.783

B2

I believe that at social events biodegradable plastic would be beneficial

0.850

0.809

B3

I believe that bio plastic will take the place of conventional polymers

0.887

0.751

B4

I believe that pollution studies of plastic should be increased

0.903

0.815

A maximum value of 0.917 was obtain for item B1 – ”I believe in biodegradable packages take away products” by Romanian students which means that students believe that bioplastic represent the future material and will replace plastic as a solution to reduce plastic waste. In exchange, the maximum value of 0.809 was obtained by Turkish students for item B2–“I believe that at social events (festival, fair, etc.) disposable biodegradable plastic would be beneficial” because they know the importance of plastic and recycling material process, but students from both countries cannot take action, or they do not have the instruments to manage the situation. 

For the variable ‘knowledge’ we select the items as in Table 21, the maximum value of 0.820 was obtained by item K2 - “Bio plastics are produced from raw materials that do not harm nature” for Romanian students, that means that they have information and knowledge about the importance of protection of the environment and recycling plastic.

Also the same item obtained the maximum value for Turkish students. Along with the activity of recycling, the students have enough information in the field of plastic waste, so they try to adapt to the new trend regarding the selection of non-polluting products.  

Table 21. Items for students’ knowledge

Item

Question

Romania

Turkey

K1

I know that the air is polluted by burning plastic

0.663

0.528

K2

Bioplastics are produced from raw materials that do not harm nature.

0.820

0.695

K3

Bioplastics are produced from raw materials that do not harm human health.

0.752

0.686

K4

I know that it is not good to throw plastic products into nature

0.743

0.219

K5

I know how to choose products that are less harmful to nature

0.707

0.569

K6

I know that I have to throw away the plastic products from the green area

0.490

0.562

K7

Bioplastic products do not cause an increase in the greenhouse gas effect

0.584

0.621

The lowest value of 0.523 was obtain by Turkish students for item K1-I know that the air is polluted by burning plastic”, and for Romanian students a value of 0.490 for item K6-”I know that I have to throw away the plastic products from the green area” which present a young generation a little bit lazy when it comes to following and observing rules.

The maximum value was obtained by both countries for Preference and Knowledge variable shows us that students are not influenced in their beliefs and culture in our country’s case; they are interested in the issue of plastic bringing  information without borders regarding the important problems of the planet; they are informed and they know everything there is around theme. The weak values for variables Knowledge, Belief and Preferences for students from both countries sustain that students in the future need more knowledge which can be handled by training, or by universities and organizations which can involve them in different activities and create an educational culture for the protection of the environment and influence their behavior and attitude with the new knowledge and information obtained.

  1. Conclusion

The proposed model starts from the cultural differences (country), hypothetically identified as an impediment and was transformed into a successful instrument by turning obstacles into opportunities through a frontal approach.

The challenge of today due to globalization sustains the importance of understanding other cultures and the importance of culture in production, business, and quality management and encourages education for new challenges.

           The study proposed a cross-cultural model regarding students‘ perception from Romania and Turkey and identified students’ preferences regarding plastic waste.

The results confirm that country, or traditional national lifestyle cannot influence the students’ preferences or knowledge in the digital era when information is without borders. The young generation is very curious and obtains everything using digital technology but they do not pay attention to the small details in their daily life maybe because they are concentrated on their education.

           The cross-model wants to highlight the preferences and knowledge of students about plastic waste and the results showed that there is no differences between students’ information and students’ knowledge about plastic waste between the two countries; also gender or specialization have no influence on the perception of bio plastic.

Another common point for students from both countries is that in a low percent they participate and attend conferences about nature protection, plastic waste, and a constant percentage of students do not participate in any conference about nature conservation or recycling materials and they are on standby, maybe because it is not specific to their field of study.

As a conclusion we can mention that Turkish students are more responsible and more active in environmental activities regarding plastic waste in comparison with Romanian students. Turkish students are more careful with recycling waste plastic, and they choose products less harmful to nature in comparison with Romanian students.

As a final conclusion university can apply the 7D’s circle as shown in Figure 6:

       Figure 6. The 7D’s for cross cultural development

  1. direction to give students information regarding the plastic waste using digital technology (digital platform for courses and lectures), small videos to understand better the phenomenon;
  2. decision to involve students in laboratory activities and research work as a practical period;
  3. documentation – working together in same research topic using good practice and change of methods and results, transfer of information between universities;
  4. diversification – harmonization with the last trends from the industry and the market, create and develop new specialization, new jobs on the market because of plastic waste;
  5. development - harmonization with the new digital technology (mobile phone, digital platforms, virtual reality, ZOOM meeting) universities can adapt the lectures and curricula in correlation with the students’ needs, share ideas and information;
  6. desire - the needs of the young generation to adapt to the new trends;
  7. durability - the open access to instruments used in and by universities can attract the young generation in research or volunteer activities.

Limitation

           The cross-cultural model was applied only to students from three universities, from two countries, Romania and Turkey. It is useful for universities to identify the students’ level of knowledge regarding different topics and obtain solutions to increase students’ interest. Also it would be interesting to apply the same model amongst academic staff and workers from industry who are involved in the technological process and plastic waste management.

 Point 11

 Although they are lifting the model from a previous study, it would have been useful for the readers that they also explain a bit the cross model in this paper.

Response 11

We apply the same  model separate for Romania and Turkey.

Point 12

 Comments on the Quality of English Language

Response 12

A professionist verify te engliish mistakes.  

Reviewer 2 Report

Comments and Suggestions for Authors

Line 62 missing a space between reference and et al.

I believe that in this document it would be extremely important to mention the different environmental and health impacts, currently known, that derive from the use and accumulation of plastics, since its various impacts on the environment and health can help to make populations aware of reducing consumption or stop using it. Mention main types of plastics, derived contaminants such as emerging contaminants BPA, 4-noniphenol, PFOS, microplastics, heavy metals and many others. It would probably fit well in section 2.

 Line 430 “importance” it's misspelled.

Line 454 have an extra space.

Author Response

Response to Reviewer 2 Comments

Dear Reviewer

First of all we want to thank you for your attention and time to review our article and for your suggestions.

Point 1-Line 62  missing a space between reference and et al.

Response 1

We correct

Point 2

I believe that in this document it would be extremely important to mention the different environmental and health impacts, currently known, that derive from the use and accumulation of plastics, since its various impacts on the environment and health can help to make populations aware of reducing consumption or stop using it.

Response 2

We change the entier paragraph

  1. Introduction

Plastic is probably the biggest challenge today, but there are problems with all types of packaging waste, so the population needs a favorable framework through which selective collection can no longer be considered an effort. Romania ranks last in the European Union in terms of packaging waste recycling, the percentage falling in 2020 to 39%, with almost 5 percentage points less than in 2019, according to the latest European statistics cited by Clean Recycle [1]. Comparatively, the EU champions in this chapter are Belgium with 79%, the Netherlands with 74% and Luxembourg with 72%. The environmental targets have increased in 2023, so that packaging waste should be recycled in a proportion of 65%. From 2025, this percentage will rise to 70% [1].

Romania is faced with the unpredictability of the legislation, with a lack of awareness and education of the population towards eco-responsible behavior, but also with the lack of infrastructure [2]. Jambeck et al.[3] and Karasik [4] mention in their research that Turkey has one of the highest volumes of both plastic and overall waste in the world, with a significant waste footprint in the Mediterranean Sea, with the largest mass of mismanaged plastic waste. Edelson et al.[5] mentioned that plastic pollution, is not only limited to its recycling and selection but also to the management of plastic and pollution which must be improved. Karasik [4] emphasized that a solution to the problem of plastic pollution would be the promotion and raising awareness of this problem, which can have a positive impact on students and in the innovation of new effective solutions. The use of plastic is one of the most pressing environmental issues facing humanity in order to reduce global warming. A large part of this waste corresponds to the food industry and their packaging.

When we use the word "plastic" we generally mean to describe the multitude of shapes and forms in which this material appears. There are seven types of plastic that vary in chemical composition, purpose, recyclability and hazardous nature. Regardless of which category of the seven types of plastic they fall into, all plastics must be recycled or reused to move towards a circular economy. We must also mention other types of plastic materials, derived contaminants, such as the emerging contaminants BPA, 4-noniphenol, PFOS, micro plastics, heavy metals and many others. Plastic must be recycled or reused for a circular economy to mitigate pollution and its impact on the planet. Most recycling programs do not accept the seventh category of plastics because they are difficult to identify and separate for recycling.

Reducing plastic consumption is extremely essential to mitigating pollution and its impact on the planet. PET plastic is mainly used as packaging for juice, water, medicine jars, household cleaning products and more. PET plastic is one of the most frequently recycled. The use of plastic is one of the most pressing environmental issues facing humanity in order to reduce global warming. Also a large part of this plastic waste corresponds to the food industry and their packaging. Therefore, it is important to know which packaging is dangerous and which safe options exist. The risks to human health and the environment associated with the use of plastic containers are huge. Aurisano et al.[6] specified, however, that the activation of a circular economy for plastics in Europe is an ambitious objective. Assessing their impact on the environment and on human health throughout the life cycle of plastic products is paramount. They identified 1,518 chemicals of concern related to plastic, replacing them with safer alternatives in support of a circular plastics economy. Verla et al.[7], however, consider that the current problem faced by researchers globally is micro plastic, as well as the toxic chemical pollution of the ecosystem and the impact of ingested micro plastic on human health.

Point 3

Mention main types of plastics, derived contaminants such as emerging contaminants BPA, 4-noniphenol, PFOS, microplastics, heavy metals and many others. It would probably fit well in section 2.

Response 3

The use of plastic is one of the most pressing environmental issues facing humanity in order to reduce global warming. A large part of this waste corresponds to the food industry and their packaging.

When we use the word "plastic" we generally mean to describe the multitude of shapes and forms in which this material appears. There are seven types of plastic that vary in chemical composition, purpose, recyclability and hazardous nature. Regardless of which category of the seven types of plastic they fall into, all plastics must be recycled or reused to move towards a circular economy. We must also mention other types of plastic materials, derived contaminants, such as the emerging contaminants BPA, 4-noniphenol, PFOS, micro plastics, heavy metals and many others. Plastic must be recycled or reused for a circular economy to mitigate pollution and its impact on the planet. Most recycling programs do not accept the seventh category of plastics because they are difficult to identify and separate for recycling.

Reducing plastic consumption is extremely essential to mitigating pollution and its impact on the planet. PET plastic is mainly used as packaging for juice, water, medicine jars, household cleaning products and more. PET plastic is one of the most frequently recycled. The use of plastic is one of the most pressing environmental issues facing humanity in order to reduce global warming. Also a large part of this plastic waste corresponds to the food industry and their packaging. Therefore, it is important to know which packaging is dangerous and which safe options exist. The risks to human health and the environment associated with the use of plastic containers are huge. Aurisano et al.[6] specified, however, that the activation of a circular economy for plastics in Europe is an ambitious objective. Assessing their impact on the environment and on human health throughout the life cycle of plastic products. They identified 1,518 chemicals of concern related to plastic, replacing them with safer alternatives in support of a circular plastics economy. Verla et al.[7], however, consider that the current problem faced by researchers globally is microplastic, as well as the toxic chemical pollution of the ecosystem and the impact of ingested microplastic on human health.

Point 4

Line 430 “importance” it's misspelled.

Response 4

We correct

Point 5

Line 454 have an extra space.

Response 5

We correct

Reviewer 3 Report

Comments and Suggestions for Authors

The work described in this paper investigates about the differences between Turkish and Romanian students, in terms of perception of plastic-waste-

The main tool used by the authors si a questionnaire, and the data have been analysed by using the SPSS software.

Authors found some differences between the two groups, but substantially assert that new generations are quite well open to selective recycling.

The authors, however, should better specify the scientific objective of the work, by clearly stating what they expect to find and which is the current gap that this work aims to bridge.

The literature review should be improved, especially about the use of compostable materials in place of plastic (e.g. https://doi.org/10.1016/j.ese.2023.100254; 10.1108/JEDT-02-2022-0118 ; 10.4081/jae.2020.1088) and plastic biodegrability (e.g. 10.18260/1-2--16014 ;  Olteanu and Gorghiu 2023).

In addition, it would be necessary to perform a cross analysis of the data, in order to identify possible interactions among parameters (e.g. between the participation to conferences and the preference for bioplastic).

Figure 2 is difficult to read

Figure 3 needs to be better explained.

As for the objective, also conclusions are unclear. Please better specify the limits of this work and the future developments (if any). It is also needed to clearly discuss about the impact of this work for society, academia and (if appropriate) for industry.

Comments on the Quality of English Language

Please also perform a comprehensive grammar check on the whole manuscript.

some examples:

effcient at Row 56

students' at Row 387
Romnaian at Row 387

there are many other typos and errors. Please also revise each paragraph, which are often difficult to read.

Author Response

Response to Reviewer 3 Comments

Dear Reviewer

First of all we want to thank you for your attention and time to review our articles and for your suggestions

Point 1

The work described in this paper investigates about the differences between Turkish and Romanian students, in terms of perception of plastic-waste.

The main tool used by the authors si a questionnaire, and the data have been analysed by using the SPSS software.

The main tool used by the author’s si a questionnaire, and the data have been analysed by using the SPSS software

Authors found some differences between the two groups, but substantially assert that new generations are quite well open to selective recycling.

Point 1

The authors, however, should better specify the scientific objective of the work, by clearly stating what they expect to find and which is the current gap that this work aims to bridge.

Response 1

Motivation of the Study

The plastic waste management problem is a topical issue, but it is perceived differently from one country to another, from individual to individual. The purpose of the present study is to identify the common points and the differences in students’ perception and the way to approach and report on this issue regarding plastic waste. The data was used for the development of a common project, FUTURE Bio, and to bring new digital technology in student’s education. The new cross-cultural model was created based on a new study having a new target group, namely students’ from Turkish and Romanian universities, and establish if the model is useful and can investigate the cross-cultural differences of students’ knowledge, believes and perception on plastic waste problems.

The objectives of the actual research were:

  1. Identify the strong and weak points of students’ information regarding plastic waste;
  2. Identify and measure students’ knowledge in plastic waste;
  3. Identify the missing information of students regarding the plastic waste;
  4. Create new tools for students to attract their interest in the field of plastic waste (VR virtual world, digital platform, short videos);
  5. Changing the habits of students regarding the use of plastic;
  6. Acquiring notions about the waste hierarchy;
  7. Knowledge of packaging materials and selective collection;
  8. How to avoid plastic in everyday life

Point 2

The literature review should be improved, especially about the use of compostable materials in place of plastic (e.g. https://doi.org/10.1016/j.ese.2023.100254; . https://doi.org/10.1108/JEDT-02-2022-0118 ; . https://doi.org/10.4081/jae.2020.1088) and plastic biodegrability (. https://doi.org/10.18260/1-2--16014 ;  Olteanu and Gorghiu 2023).

Response 2

Complete with the suggest articles

Fadhullah et al. [30], Henderson and Dumbili [31] in their research, found that for many young Nigerians perception of plastic waste from social point of view, it’s consider to be cool using trash cans or to recycle, which implying the need for individual responsibility, all these as a sign of modernity and increased social status. Also GherheÅŸ et al. [32] investigates the student’s perceptions regarding the plastic waste and results presents that majority of students still need to be familiarized with through different campaigns, trainings, courses, etc.

Rosario and Dell [33] were concerned about the environment and the importance of sustainable materials in industry. They consider materials and laboratory courses allow students to test the biodegradability of plastic materials so that students to understand the new materials. Thus the Implementation of biodegradable testing in a curriculum provides active learning through practice tests and encourages students to engage in lifelong learning continue to develop their knowledge of emerging materials.

Currently, the recycling of metal, wood, paper and especially all cardboard packaging has improved significantly, on the other hand for plastic materials their recycling and removal from the market is not yet resolved. The European Union has implemented various regulations regarding packaging and packaging waste recycling and market implementation of eco-sustainable packaging. Rossi et al. [34] have developed an innovative and sustainable composite material for packaging a new eco friendly material based on the combination of natural biodegradable fibers and biopolymers consisting made of a material of straw and biodegradable plastic. The authors present the results of the new material showing a good match with the characteristics of current polymers, suggesting that this material can be used as a potential substitute in packaging applications. Also, Fiorineschiet et al.[35] in their paper presents the application of a systematic engineering design procedure also adapted for eco friendly production of compostable straw fiber packaging and bioplastic but in the field of viticulture (the obtained boxes are intended to be used for wine bottles ).Olteanu and Gorghiu [36] argue that scientific actions through investigations, experiments and research are important in attracting the younger generation to science. Important subjects are promoted to students in this sense, the subject of biodegradability of plastic materials, in various approaches, addressing the problems of our day, answering scientific questions or trying to do so. Olteanu and Gheorgiu [36] emphasized that the involvement of students in research leads to an increase in the interest of the young generation in science. The results obtained through specific approaches to STEM (Science, Technology, Engineering, Math) education led students to become aware of this sensitive issue. Given the assessment of students' interest in science after the implementation of the scientific actions their power confidence in science, being ready to participate in or benefit from collaborative scientific projects the support of their family who believe that the understanding and knowledge of science is useful to the whole life. Also when assessing students' interest in science, students provided positive feedback related to teachers' ability.

The transition to a circular and sustainable economy can be viewed from a socio-technical point of view response to environmental impact. Bio plastics, typically plastics made from bio-based polymers, should contribute more sustainable plastic life cycles as part of a circular economy. Ali et al.[37] in their work carry out a review that highlights the harmful effects of fossil-based plastic on environment and human health, as well as the massive need for green alternatives such as biodegradable ones bio plastics.

The use of new types of bio plastics derived from renewable resources and choosing the appropriate end-of-life option may be the right direction to ensure the sustainability of bio plastic production. At the same time, clear regulations are required and financial incentives to scale up with application in the market having a truly lasting impact.

References

Rosario, L., & Dell, E. (2010, June), Biodegradability Of Plastics Testing In An Undergraduate Materials Laboratory Course Paper presented at 2010 Annual Conference & Exposition, Louisville, Kentucky. 10.18260/1-2—16014

Rossi, G. ., Conti, L. ., Fiorineschi , L. ., Marvasi, M., Monti, M., Rotini, F., Togni, M. and Barbari, M. (2020) “A new eco-friendly packaging material made of straw and bioplastic”, Journal of Agricultural Engineering, 51(4), pp. 185–191. doi: 10.4081/jae.2020.1088.

Olteanu, R. L., & Gorghiu, G. (2023). Increasing the students’ interest in science by implementing a science action dedicated to plastics biodegradability. In V. Lamanauskas (Ed.), Science and technology education: New developments and Innovations. Proceedings of the 5th International Baltic Symposium on Science and Technology Education (BalticSTE2023) (pp. 162-172). Scientia Socialis Press. https://doi.org/10.33225/BalticSTE/2023.162

Point 3

In addition, it would be necessary to perform a cross analysis of the data, in order to identify possible interactions among parameters (e.g. between the participation to conferences and the preference for bioplastic).

Response 3

If we analyze the correlation between students preference for bio plastic and their conferences attendance from Table 3 we can observed that from both countries a low percent of 15.11% percent are agree to participate to conference and 20.20 % percent are not interesting on topic. A 42.29 % percent from students are not decided yet regarding the bio plastic subject.

Table 3. Distributoon betweeen students attendence to conferences and preference to use bioplastic

I prefer bioplastic products, even if they are expensive.

Totally Appropriate

Appropriate

Somewhat Appropriate

Not Appropriate

Not At All Appropriate

Total

Have you attended a conference on nature conservation before?

 Yes

26

63

99

35

15

238

 No

45

74

150

44

38

351

Total

71

137

     249

79

53

589

Even universities from both countries organized and host environmental activities regarding plastic and bio plastic only 5.72% percent from students participate even the new material bio plastic it is expensive. So culture don’t influence the students participation or attending to different events organized by universities.

Table 4.  Distribution  between students participation in activities organized within the university and preference to bioplastic

I prefer bioplastic products, even if they are expensive.

Totally Appropriate

Appropriate

Somewhat Appropriate

Not Appropriate

Not At All Appropriate

Total

Did you take part in environmental activities organized within the university?

Yes

10

24

35

12

5

86

No

61

113

214

67

48

503

Total

71

137

249

79

53

589

In conclusion the student’s culture not affected their attitude regarding the participation and attendance in extracurricular activities but for universities is a good signal to improve and to identify new opportunities to attract and involve the new generation. 

Point 3

Figure 2 is difficult to read

Response 3

Figure 2. Students’ preferences and their contribution to recycling process.

Using the tree analyze it was possible to see the connection between the perception of using biodegradable plastic of students from both countries like in Figure 2.

Even they contribute to recycling process the price influence their behavior. The students’ from both countries in 12.1 % percent are totally agreed with the increasing price of bio plastic and 78.9 % are somewhat appropriate with the higher price.

 A percent of 35.5 % are Turkish students which are somehow agree and 8.8 % are not agree, in comparison Romanian students’ are somehow agree in 16% percent with the increasing price and 18.7 % consider that it is not an appropriate measure taken in consideration the social life living standard from country.

Point 4

Figure 3 needs to be better explained.

Response4

Figure 3. Classification results for students’ participation in activities and campaigns about bioplastic

     Figure 3 complete the cross cultural model regarding students’participation to different activities organized by universities or society even the price of bio plastic it is expensive.  

Students’from both countries in 55.5 % percent participate to activities and even 12.1 % are totally agree and 23.3 % somewhat agree with the increasing price of bio plastic maybe  because they take in consideration that are the first products using the bio plastic material on market.

Those who are on the sidelines and are stil undecided to participate to campaigns and pther extra activites are in 46.5 % percent. In conclusion universities as nurseries of future specialists must to find solution and ways to raise  awareness and attract students

Point 5

As for the objective, also conclusions are unclear. Please better specify the limits of this work and the future developments (if any). It is also needed to clearly discuss about the impact of this work for society, academia and (if appropriate) for industry.

Response 5

The proposed model starts from the cultural differences (country), hypothetically identified as an impediment and were transformed in a successful instrument by turning obstacles into opportunities through a frontal approach.

The challenge of today due to globalization sustains the importance of understanding other culture and its importance in production, business, and quality management and encourages education for a new provocation.

The study proposed a cross cultural model regarding students ‘perception from Romania and Turkey two different countries and identify students’ preferences upon plastic waste.

The results confirm that country, or traditional national life style cannot influence the student’s preferences or knowledge in the digital era when information’s are without borders. The young generation it is very curious and obtains everything using digital technology but they didn’t pay attention to small detail element from their daily life maybe because they are concentrated to their education.

           The cross model wants to highlight the preferences and knowledge of students about plastic waste and the results showed that there is no differences between students’ information and students’ knowledge about plastic waste between the two countries, also gender or specialization has no influence on the perception of bio plastic.

Another common point for students’ from both countries it is that in a low percent they participate and attend conferences about nature protection, waste plastic and a constant percent from students’ didn’t participate to any conference about nature conservation or recycling materials and they are staying in standby, maybe because it is not specific to their specialization field.

As a conclusion we can mention that Turkish students are more responsible and more active in environmental activities regarding plastic waste in comparison with Romanian’s students. Turkish students are more carefully on recycling waste plastic, and they choose product less harmful to nature in comparison with Romanian students’.

As a final conclusion university can take apply the 7D’s circle like in Figure 6:

       Figure  6. The 7D’s for cross cultural development

  1. direction to give students information regarding the plastic waste using the digital technology (digital platform for courses and lectures), small videos to understand better the phenomen;
  2. decision to involve students in laboratories activities and research work as a practical period;
  3. documentation – working together into same topic research using good practice and change of methods and results, transfer of informtion between universities;

4.diversification – harmonization with the last trends from industry and market, create and developt new specialization, new jobs on market because of plsatic waste;

  1. development- harmonization with the new digital technology (mobile phone, digital platforms, virtual reality, ZOOM meeting) universities can adapt the lectures and curriculas in function of students needs, share ideas and information;
  2. desire-the needs of young generation to adapt with new trends;
  3. durability- the open access to instruments used in and by universities can attract young generation in research or volunteer activities.

Limitation

      The cross cultural model was applied only to students from three universities, from two countries Romania and Turkey. It is useful for universities to identify the students level of knowledges regarding different topics and obtain solution to increase students’ interest. Also will be interesting to apply the same model between academic staff and workers from industry who are involved in technologycal process and plastic waste  management

Point 6

Comments on the Quality of English Language

Response 6

An english professionist verify and correct the article.

Reviewer 4 Report

Comments and Suggestions for Authors

The title " A Cross-cultural Analysis for Plastic Waste Perception of Stu- 2 dents from Romania and Turkey " does not show clearly the model developed. It is not clearly understood its purpose. For example, in the page 3 " There are still unsolved problems regarding plastic waste, one factor would be that 102 the community still lacks awareness about this issue. Students bring water bottles to 103 campus, but still buy drinks in plastic packaging for bottle supplies. Most already know 104 the effects of plastic waste and show a desire to reduce plastic bottle waste."… This is fuzzy writing.

This research does not have working hypotheses, which are essential in any research. The way the questionnaire was carried out is not known. The discussion does not follow the logic of the presentation of the research model. The findings do not present research findings. The work also presents no contributions, limitations (very vague) and recommendations for future research.

In the "Results" section this is descriptive statistics in the Table 1-Table 8. Tables should all be simplified by following the rules of a research with high quality. The authors should create an academic research model. Please revise figure 2 and figure in the page 11-12.

This research also presents more difficulties than those referred to:
1) Lack a structured introduction. They do not present a framework based on the literature, and not refer to the theory of support the research, gaps of research, statistical methodologies, the main contributions, and the sequence of work;

2) The text does not specify the research objectives;

3) The structure of "literature review and research hypotheses" should follow the logic of the research model, which is not done. Although the authors refer to these hypotheses in the title are not displayed in the research;
4) The "Results" section should present the results in the logic of the research model and hypotheses of work (not presented). Tables should all be simplified by following the rules of a research. The authors should create acronyms for variables.  Table 9- Table 11 is not clear to me.
5) The findings are a confusion; The discussion is where the authors have invested most in this work but does not respect the research model in its logical construction. The argument made is very partial and no logical relation to the purpose of the work (given the model). The literature was very little used in the discussion;

The most important point of this work to focuses on the diagram of the research model. Without this model would be incomprehensible. Also focuses on applied statistics with many weaknesses in the tests and how the data are interpreted (not developed this point because previous work shows enormous deficiencies that do not give credibility to the calculated data).

There would be many other critics and comments to make. However, in the face a lot of difficulties do not delve further analysis. This work has profound problems of form, methodology and substance. It would be the substances of issues that we should focus the critical comments which are prevented by primary issues.

All these comments are to help improve the research effort. Good work and good luck.

Comments on the Quality of English Language

The English writing is very weak and must be improved for meeting the quality standard.

Author Response

Response to Reviewer 4 Comments

Dear Reviewer

First of all we want to thank you for your attention and time to review our articles and for your suggestions

Point 1

The title " A Cross-cultural Analysis for Plastic Waste Perception of Students from Romania and Turkey " does not show clearly the model developed. It is not clearly understood its purpose.

Response 1

We change and introduce  a new direction creating separate two models, one for Raomania and one for Turkey and then make a comparison.

Motivation of the Study

The plastic waste management problem is a topical issue, but it is perceived differently from one country to another, from individual to individual. The purpose of the present study is to identify the common points and the differences in students’ perception and the way to approach and report on this issue regarding plastic waste. The data was used for the development of a common project, FUTURE Bio, and to bring new digital technology in student’s education. The new cross-cultural model was created based on a new study having a new target group, namely students’ from Turkish and Romanian universities, and establish if the model is useful and can investigate the cross-cultural differences of students’ knowledge, believes and perception on plastic waste problems.

The objectives of the actual research were:

  1. Identify the strong and weak points of students’ information regarding plastic waste;
  2. Identify and measure students’ knowledge in plastic waste;
  3. Identify the missing information of students regarding the plastic waste;
  4. Create new tools for students to attract their interest in the field of plastic waste (VR virtual world, digital platform, short videos);
  5. Changing the habits of students regarding the use of plastic;
  6. Acquiring notions about the waste hierarchy;
  7. Knowledge of packaging materials and selective collection;
  8. How to avoid plastic in everyday life

The structural model for Romania is given in Figure 4 where we identify the connection and relationship between variables and indicate that while the Knowledge has a statistically significant effect on Preferences with the coefficient of 0.427.

Figure 4. PLS Path Results for Romania

Figure 5. PLS Path Results for Turkish students’

Point 2

For example, in the page 3 " There are still unsolved problems regarding plastic waste, one factor would be that the community still lacks awareness about this issue. Students bring water bottles to  campus, but still buy drinks in plastic packaging for bottle supplies. Most already know  the effects of plastic waste and show a desire to reduce plastic bottle waste."… This is fuzzy writing.

Response 2

We change in text and delete

Point 3

This research does not have working hypotheses, which are essential in any research. The way the questionnaire was carried out is not known. The discussion does not follow the logic of the presentation of the research model. The findings do not present research findings.  The work also presents no contributions, limitations (very vague) and recommendations for future research.

Response3

We change and improve the  entire text in  a specific order suggested by Reviewer.

  1. Research Methodology

3.1. Materials and Methods 

A total of 589 students, were involved in a choice experiment during which a specially designed questionnaire through face-to-face interviews and online was applied between November – December 2022 in two countries with different culture, between students’ from North Center University of Baia Mare, branch of Technical University of Cluj Napoca, Romania and Pamukkale University from Denizli and Kirkareli University both universities from Turkey.  

The survey was applied on online platform using Google Drive forms. The survey had 29 questions from which 20 questions that tested students' knowledge, believes and preferences in using the bio plastic, recycling process about waste plastic and environmental pollution and 5 questions regarding the individual characteristic and 4 questions regarding student’s attitude. The following variables were taken in consideration for the cross cultural model like in Table 1:

I—Individual characteristics (age, gender, grade of education level, university, faculty, specialization, country);

A- Attitude-participation to conferences, activities organized by university, volunteer activity.

K—Knowledge’s regarding bio plastic and impact of plastic in their daily life;

P - Preferences if they prefer plastic, or new biodegradable plastic, how they choose the products, if they take care of type the material, if they agree with extra money for new bio plastic;

B – Believes –if they have information about the plastic waste, plastic pollution, if they believe in replacement of plastic.

To measure student’s knowledge’s and preferences on plastic waste, a similar scale used Boca and Saraçli [46] in their research about students’ awareness were used to obtain realistic information. The Likert scale was ranging from 1 to 5, where 1- represent ‘’ Not at all appropriate ‘’ and 5 – ‘’ Totally appropriate’’.

The statistical instruments used for data analyze was the SPSS 25 software package and the SMART PLS4 program. The research between the two countries has been done for a better understanding of students’ believes regarding sustainable environment and to examine their preferences on specific topic plastic and bio plastic.

Table 1.  Survey structure and variables taken in consideration

KNOWLEDGE

K1

I know that the air is polluted by burning plastic

K2

Bioplastics are produced from raw materials that do not harm nature.

K3

Bioplastics are produced from raw materials that do not harm human health.

K4

I know that it is not good to throw plastic products into nature

K5

I know how to choose products that are less harmful to nature

K6

I know that I have to throw away the plastic products from the green area

K7

Bioplastic products do not cause an increase in the greenhouse gas effect

PREFERENCE

P1

I prefer to use mesh/cloth/paper bags instead of using disposable bags while shopping.

P2

I prefer to use bioplastic bags for my grocery shopping.

P3

I prefer bioplastic products, even if they are expensive

P4

I prefer bioplastic products because they degrade earlier in nature.

P5

I prefer the products obtained from the bioplastics industry because they are renewable.

P6

I prefer bioplastic products because they do not harm the nature when they decompose.

P7

I prefer bioplastic products as they do not harm human health when degraded.

P8

I prefer to use bioplastic products in the kitchenware.

P9

I prefer not to buy products with nylon additives

BELIEVES

B1

I believe in biodegradable packages take away products.

B2

I believe that at social  events biodegradable plastic would be beneficial

B3

I believe that bio plastic will take the place of conventional polymers

B4

I believe that pollution studies of plastic should be increased

This study provides a framework on which the authors build a more in-depth examination of the factors that influence students’ perception, knowledge’s, believes and attitude towards plastic waste, thus a research model like as show in Figure 1 were used.

Figure 1. Research model for Students’ Cross Cultural Model regarding plastic waste

     The hypotheses tested on the perception attitude of students in the present study are:

H1: Knowledge effects Preference regarding attitude of Romanian students towards waste plastic;

H2: Believes effects Preference regarding attitude of Romanian students towards waste plastic;

H3: Knowledge effects Preference regarding attitude of Turkish students towards waste plastic;

H4: Believes effects Preference regarding attitude of Turkish students towards waste plastic.

As a final results of study will be a cross-cultural model which will help the universities and academic staff to adapt the curricula and academic activities to encourage and to prepare common project and changes of good practice for a better cooperation without borders. Because the cross cultural model it is referring to student’s culture the authors take in consideration the country as a cultural factor of influence.

Point 4

In the "Results" section this is descriptive statistics in the Table 1-Table 8. Tables should all be simplified by following the rules of a research with high quality. The authors should create an academic research model. Please revise figure 2 and figure in the page 11-12.

Response 4

We create the model of research

This study provides a framework on which the authors build a more in-depth examination of the factors that influence students’ perception, knowledge, beliefs and attitude towards plastic waste, thus a research model as shown in Figure 1 was used.

Figure 1. Research model for Students’ Cross Cultural Model regarding plastic waste

We change also the table format

Table 2. Student distribution according to field of study

 Question                                              Specialization  

Romania

Turkey

Cumulative  percent

Which faculty do you study at

Industrial engineering

0

20

3.39

Faculty of Sciences

226

60

48.56

Denizli Technical Institute

0

 59

  10.02

Mechatronic Faculty

0

 51

  8.66

Textile Faculty

0

45

7.64

Faculty of Engineering

69

27

 16.30

Medicine

0

32

 5.43

Total

295

294

100

We change the figures

Figure  2. Students’ preferences and their contribution to the recycling process

Figure  3. Classification results for students’ participation in activities and campaigns about bioplastic

Point 5

This research also presents more difficulties than those referred to:
1) Lack a structured introduction. They do not present a framework based on the literature, and not refer to the theory of support the research, gaps of research, statistical methodologies, the main contributions, and the sequence of work;

Response 5

2) The text does not specify the research objectives;

Motivation of the Study

The plastic waste management problem is a topical issue, but it is perceived differently from one country to another, from individual to individual, the purpose of the present study is to identify the common points and the differences in students’ perception and the way to approach and report on this issue regarding plastic waste. The data were used for the development of a common project FUTURE Bio and bring new digital technology in student’s education. The new cross cultural model were created on base of a new study having a new target students’ from Turkey and Romania universities and establish if the model is useful and can investigate the cross cultural differences of students’ knowledge, believes and perception on plastic waste problems.

The objectives of actual research were:

  1. Identify the strong and weak points of students’ information regarding plastic waste;
  2. Identify and measure students knowledge’s’ in plastic waste;
  3. Identify the missing information of students regarding the plastic waste;
  4. Create new tools for students’ to attract their interest in plastic waste field (VR virtual world, digital platform, short videos).
  5. Changing the habits of students o=to use plastic;
  6. Acquiring notions about the waste hierarchy;
  7. Knowledge of packaging materials and selective collection;
  8. How to avoid plastic in everyday life.

Point 6

3) The structure of "literature review and research hypotheses" should follow the logic of the research model, which is not done. Although the authors refer to these hypotheses in the title are not displayed in the research;

Response 6

This study provides a framework on which the authors build a more in-depth examination of the factors that influence students’ perception, knowledge’s, believes and attitude towards plastic waste, thus a research model like as show in Figure 1 were used.

Figure 1. Research model for Students’ Cross Cultural Model regarding plastic waste

     The hypotheses tested on the perception attitude of students in the present study are:

H1: Knowledge effects Preference regarding attitude of Romanian students towards waste plastic;

H2: Believes effects Preference regarding attitude of Romanian students towards waste plastic;

H3: Knowledge effects Preference regarding attitude of Turkish students towards waste plastic;

H4: Believes effects Preference regarding attitude of Turkish students towards waste plastic.

Point 7

4) The "Results" section should present the results in the logic of the research model and hypotheses of work (not presented). Tables should all be simplified by following the rules of a research. The authors should create acronyms for variables.  Table 9- Table 11 is not clear to me.

Response 7

Point 8

5) The findings are a confusion; The discussion is where the authors have invested most in this work but does not respect the research model in its logical construction. The argument made is very partial and no logical relation to the purpose of the work (given the model). The literature was very little used in the discussion;

Response 8

Point 9

The most important point of this work to focuses on the diagram of the research model. Without this model would be incomprehensible. Also focuses on applied statistics with many weaknesses in the tests and how the data are interpreted (not developed this point because previous work shows enormous deficiencies that do not give credibility to the calculated data).

Response 9

5.3. A Cross Cultural Model for Students Perception Regarding Plastic Waste

We took in consideration the data base after applying surveys to 589 students from Romania and Turkey universities. Following Hair et al.[48], Hair et al.[49], Ringle et al.[51] and Sarstedt [52] researches and using the same program Smart PLS 4 was possible to establish the cross model.

We present the SMART PLS 4 program solution, and also the cross cultural model taking in consideration the Romanians and Turkish students’ knowledge, perception and students’ believes and compare the results. To model the relations among sub-dimensions and compare Romania and Turkey we applied PLS-SEM.

The results of Factor Loadings, Cronbach's Alpha, CR and AVE values, Fornell-Larcker [53] criterion findings HTMT criteria findings, parameter estimates and t statistics of the PLS model, and findings on effect sizes (f 2) are given in Tables 9 - 13 for Romania  and Tables 14-18 for Turkey.

Structural model for Romania it is given in Figure 4 and for Turkey in Figure 5.

There are three criteria to ensure convergent validity in the PLS model.

  1. first, each standard factor loading of latent variables is greater than 0.5 and statistically significant;
  2. second: Structural Reliability (CR) and Cronbach's Alpha (CA) values for each structure should be greater than 0.7
  3. the third criterion is: The Average Variance Explained (AVE) value should be higher than 0.5 (Fornel & Lacker [53], Hair et al.[54]).

  1. Cross cultural model for Romanian students regarding plastic waste

Factor loadings, CA, CR, AVE values of affective, behavioral and cognitive dimensions are given in Table 9 for Romanian students’. As we can see the latent variables values are valid and accepted.

                         Table 9. Factor Loadings, Cronbach's Alpha, CR and AVE values for Romanian students’

Item/Dimension

Factor Loadings

CR

AVE

K

K1

0.663

0.809

0.860

0.473

K2

0.820

K3

0.752

K4

0.743

K5

0.707

K6

0.490

K7

0.584

B

B1

0.903

0.912

0.914

0.792

B2

0.917

B3

0.850

B4

0.887

P

P1

0.670

0.894

0.938

0.548

P2

0.756

P3

0.580

P4

0.854

P5

0.850

P6

0.834

P7

0.855

P8

0.612

P9

 0.569

In Table 10, the discriminant validity of the measurement model (Fornell-Larcker criterion)[53] values for all factors are given. The discriminant validity of the model is checked by comparing the square root of the AVE value for each construct with the correlations between the constructs. Here, it is said that the discriminant validity of the model is ensured if the square root values of the AVE are large [51].

                              Table 10. Fornell-Larcker criterion findings for Romanian students’

B

K

P

B

0,890

K

0,700

0,688

 P

 0,844

 0,809

0,740

When examining the Fornell-Lacker table, the diagonal values ​​represent the square root of the AVE values ​​for each factor, and the off-diagonal values ​​represent the correlation coefficients between the factors. As an alternative to the Fornell-Lacker criterion, the HTMT criterion represents the geometric mean of factor correlations which are given in Table 11. In conclusion the cross model for Romania it’s available.

                               Table 11. HTMT criteria findings for Romania.

B

K

P

B

K

0,797

P

0,895

0,962

Structural model for Romania it is given in Figure 4 identify the connection and relationship between variables and indicate that while the Knowledge has a statistically significant effect on Preferences with the coefficient of 0.427, Believes has a statistically significant effect on Preferences with the coefficient of 0.545. Knowledge (K) and Believes (B) explain the 80.5% of Preferences (P).

Figure 4. PLS Path Results For Romania

Also while K2 “Bio plastics are produced from raw materials that do not harm nature.’’ has the highest effect within Knowledge with the coefficient of 0.820, B1 “I believe in biodegradable packages take away products’’ has the highest effect within Believes with the coefficient of 0.917 and P7 “I prefer bio plastic products as they do not harm human health when degraded’’ has the highest effect within Preferences with the coefficient of 0.855. Also it can be seen from Table 12, that Hypothesis H1 and H2 for Romania are accepted and path coefficients for the model are statistically significant.

Table 12. Parameter estimates and t statistics of the PLS model for Romanian students ‘

Hypothesis

Relationship

Parameters (β)

t- statistics

p- values

Decision

H1

B  P

0.545

14.627

0.0001*

Accepted

H2

K  P

0.425

10.165

0.0001*

Accepted

*p<0.01

From Table 13 we can observed the effect sizes evaluated as >=0.02 low, >= 0.15 medium, >= 0.35 strong (Hair et al., 1998). If the VIF value is equal to or greater than 5, then it is known that there is a multi co-linearity problem (Henseler et al.[55]). Since VIF values ​​are less than 5, it can be said that there is no multi co-linearity problem. VIF values of this study are between 1.343 and 4.640.

                Table 13. Findings on effect sizes (f 2) for Romania.

Relationship

f 2 Values

Effect Size

B  P

0.779

Strong

K  P

0.478

Strong

  1. Cross cultural model for Turkish students regarding plastic waste

  Because the proposed model it is about cross culture we followed similar steps for Turkey. Factor loadings, CA, CR, AVE values of affective, behavioral and cognitive dimensions are given in Table 14 for Turkish students’. As we can see the latent variables values are valid and accepted following the standard values mention before.

                                    Table 14. Factor Loadings, Cronbach's Alpha, CR and AVE values for Turkish students’

Item/Dimension

Factor Loadings

CR

AVE

K

K1

0,528

0,644

0,762

0,329

K2

0,695

K3

0,686

K4

0,219

K5

0,569

K6

0,562

K7

0,621

B

B1

0,815

0,799

0,869

0,624

B2

0,783

B3

0,809

B4

0,751

P

P1

0,562

0,845

0,879

0,457

P2

0,526

P3

0,488

P4

0,810

P5

0,774

P6

0,831

P7

0,823

P8

0,591

P9

0,561

In Table 15, the discriminant validity of the measurement model (Fornell-Larcker criterion) values for all factors are given again the discriminant validity of the model is checked by comparing the square root of the AVE value for each construct with the correlations between the constructs. Here, it is said that the discriminant validity of the model is ensured if the square root values of the AVE are large [50].

                        Table 15. Fornell-Larcker criterion findings for Turkish students’

B

K

P

B

0,790

K

0,517

0,574

P

0,650

0,690

0,676

The alternative to the Fornell-Lacker criterion, for Turkish students’ for the HTMT criterion are given in Table 16.

                                Table 16. HTMT criteria findings for Turkish students’

B

K

P

B

K

0,707

P

0,751

0,912

Structural models for Turkey are also given in Figure 5 which indicate that while the Knowledge has a statistically significant effect on Preferences with the coefficient of 0.483, Believes has a statistically significant effect on Preferences with the coefficient of 0.400.

Figure 5. PLS Path Results For Turkish students’

Knowledge (K) and Believes (B) explain the 59.3% of Preferences (P). similarly with the Romanian model given in Figure 5, while K2 “Bio plastics are produced from raw materials that do not harm nature.’’ has the highest effect within Knowledge with the coefficient of 0.695, B1 “I believe in biodegradable packages take away products.’’ has the highest effect within Believes with the coefficient of 0.815 and P6 “I prefer bio plastic products because they do not harm the nature when they decompose.’’ has the highest effect within Preferences with the coefficient of 0.831. It can be seen from Table 17, that Hypothesis H3 and H4 for Turkey are accepted and path coefficients for the model are statistically significant.

Table 17. Parameter estimates and t statistics of the PLS model for Turkish students’

Hypothesis

Relationship

Parameters (β)

t- statistics

p- values

Decision

H3

B  P

0.400

8.279

0.0001*

Accepted

H4

K  P

0.483

10.286

0.0001*

Accepted

*p<0.01

Since VIF values ​​are all less than 5 like Romanian results, it can be said that there is no multi co linearity problem. VIF values of this study are between 1.052 and 3.497 (Table 18).

                                 Table 18. Findings on effect sizes (f 2) for Turkish students’

Relationship

f 2 Values

Effect Size

B  P

0.289

Strong

K  P

0.420

Strong

In conclusion results of PLS SEM indicate that both Knowledge (K) and Believes (B) have significant effect on students Preferences (P) for Romania and Turkey. Even some of the criteria for PLS-SEM model for Turkey do not meet, because they are close to critical limits and to evaluate the same model and compare two countries findings are given in related tables

Point 10

There would be many other critics and comments to make. However, in the face a lot of difficulties do not delve further analysis. This work has profound problems of form, methodology and substance. It would be the substances of issues that we should focus the critical comments which are prevented by primary issues

Response 10

 We consider that the pertinent suggested changes improve the article.

All these comments are to help improve the research effort. Good work and good luck.

 Point 11

Comments on the Quality of English Language

The English writing is very weak and must be improved for meeting the quality standard.

Response 11

Wthe articke was verify by an engish professionist

Round 2

Reviewer 3 Report

Comments and Suggestions for Authors

The authors improved the paper in this new version, and resolved quite well most of the doubts I rised in the precedent submission.

Please consider to change "limitation" into "limits of the work" and to move it within the discussion section. Please carefully discuss about the actual limits of the work, not only in terms of the number of considered universities and what can be interesting in your opinion. Instead, comprehensively explain the limitations of the adopted methodology, from a statistical point of view (i.e. the statistical power, the roboustness of the analysis, and if other approaches can be used in place of that considered by the authors).

Comments on the Quality of English Language

The paper is quite readable, but I suggest a comprehensive proofreading session performed  by a native English.

Author Response

Response to Reviewer 3 Comments

Dear Reviewer

First of all we want to thank you for your attention and time to review our articles and for your suggestions

Point 1

Please consider to change "limitation" into "limits of the work" and to move it within the discussion section. Please carefully discuss about the actual limits of the work, not only in terms of the number of considered universities and what can be interesting in your opinion. Instead, comprehensively explain the limitations of the adopted methodology, from a statistical point of view (i.e. the statistical power, the roboustness of the analysis, and if other approaches can be used in place of that considered by the authors).

Response 1

Limitations of the work

      This study has some limitations. First of all, the construction of the model of variables taken into consideration in the evaluation of knowledge about plastic waste is not comprehensive enough. Currently, there is no unified system of indicators, and different indicators can lead to different results; therefore, the results here should be further objectively validated. Second, plastic waste is influenced by many factors, and the dominant factors vary. Third, due to data availability limitations, the duration of this study is relatively short. Fourth, more comprehensive research is needed, starting from the characteristics of country, city university; researchers should select cities, universities with similar development characteristics for comparative studies. Because of the respondents' cultural backgrounds or perspectives on certain phenomena, this could affect the legitimacy of the study in using the model. It should be noted that the research problem has been stated and the data collection process has been carried out properly. Future research can use our study as a starting point to track changes over time, expand the study area, and further examine academic staff or industry employees.

The cross-cultural model was applied only to students from three universities, from two countries, Romania and Turkey. It is useful for universities to identify the students’ level of knowledge regarding different topics and obtain solutions to increase students’ interest. Also it would be interesting to apply the same model amongst academic staff and workers from industry who are involved in the technological process and plastic waste management.

From the statistical point, even there are other methods to determine the statistically significant variables on the categorical dependent variable, to determine the effective factors on student’s preferences about bioplastic and to give the results more visually, in this study we applied CHAID (Chi-Squared Automatic Interaction Detection) analysis. On the other hand the relationships among the dimentions can be modelled by either covariance-based SEM (CB-SEM) or partial least squares SEM (PLS-SEM). Because PLS-SEM is primarily used to develop theories in exploratory research in this study we preferred this approach. Sample size of this study (295 for Romania, 294 for Turkey and 589 in total) is large enough to apply SEM at an acceptable power (min. 80%) and all the statistical analysis are concluded at 95% confidence level.

Point 2

The paper is quite readable, but I suggest a comprehensive proofreading session performed  by a native English.

 Response 2

We  turned to a professional for English verification.. 

Reviewer 4 Report

Comments and Suggestions for Authors

The artcile replies to the reviwers' comments.  After reviewing the artilce, I agree that accept after minor English editing.

Comments on the Quality of English Language

Minor editing of English language required

Author Response

Response to Reviewer 4

Dear Reviewer

First of all we want to thank you for your attention and time to review our articles and for your suggestions

Point 1

The article replies to the reviwers' comments.  After reviewing the article, I agree that accept after minor English editing.

Minor editing of English language required

Response 1

We fix and change the english mistakes.
